# DSR: Dynamical Surface Representation as Implicit Neural Networks for Protein

**Daiwen Sun**[1], **He Huang**[2,1], **Yao Li**[2,3], **Xinqi Gong**[1,2]*, **Qiwei Ye**[2]*

[1]Institute for Mathematical Sciences, School of Mathematics,
Renmin University of China, Beijing, China
[2]Beijing Academy of Artificial Intelligence, Beijing, China
[3]School of Life Sciences, Tsinghua University, Beijing, China
sundaiwen@ruc.edu.cn, huanghe@baai.ac.cn, liyao17@tsinghua.org.cn,
xinqigong@ruc.edu.cn, qwye@baai.ac.cn

## Abstract

We propose a novel neural network-based approach to modeling protein dynamics using an implicit representation of a protein's surface in 3D and time. Our method utilizes the zero-level set of signed distance functions (SDFs) to represent protein surfaces, enabling temporally and spatially continuous representations of protein dynamics. Our experimental results demonstrate that our model accurately captures protein dynamic trajectories and can interpolate and extrapolate in 3D and time. Importantly, this is the first study to introduce this method and successfully model large-scale protein dynamics. This approach offers a promising alternative to current methods, overcoming the limitations of first-principles-based and deep learning methods, and provides a more scalable and efficient approach to modeling protein dynamics. Additionally, our surface representation approach simplifies calculations and allows identifying movement trends and amplitudes of protein domains, making it a useful tool for protein dynamics research. Codes are available at https://github.com/Sundw-818/DSR, and we have a project webpage that shows some video results, https://sundw-818.github.io/DSR/.

## 1  Introduction

Proteins, which are made up of chains of amino acids, are essential molecules in living organisms. Protein with specific three-dimensional structures are responsible for a wide range of biological processes such as enzymatic reactions, transport of molecules, and structural support. However, proteins are not static objects in vivo, but constantly in motion, with individual amino acids vibrating, rotating, and shifting in response to the change of environment. The biological functions of proteins are fulfilled by their dynamical behaviors, and researchers study the dynamics and functions of proteins by molecular dynamics(MD) simulations[1, 2].

First-principles molecular dynamics use the forces calculated from the electronic states by solving the Schrödinger equation (namely, density functional theory) to compute the coordinate trajectories[3], with programs such as VASP[4] and GPAW[5]. However, the computational expense of MD typically increases exponentially in relation to the number of electronic degrees of freedom. In order to reduce the computational cost, empirical force field (classical force field) models with simplified functional forms are employed in MD simulation, such as AMBER[6, 7], CHARMM[8, 9], and GROMOS[10–12], which are designed to capture the bonded and non-bonded interactions between atoms. Besides, machine learning techniques can be employed to fit force fields or potential energy surfaces[13–16]. These force field-based methods are much faster than ab initio methods without exhaustive QM

---

*Correspondence to: Xinqi Gong<xinqigong@ruc.edu.cn>, Qiwei Ye <qwye@baai.ac.cn>.

37th Conference on Neural Information Processing Systems (NeurIPS 2023).

calculations. However, the integration time step in MD cannot be too large to resolve the fastest motion in the system, for example the vibration of hydrogen-involved bonds whose timescales are as short as a few femtoseconds. This restricts MD from having more speed advantages. To address this problem, coarse-grained methods are being developed to perform molecular dynamics simulations faster and on larger systems by eliminating some unimportant degrees of freedom[17–20]. Fast as it is, coarse-grained simulations are only able to reproduce molecular properties at low resolution, such as $Rg^2$, sacrificing the all-atom details. Given this perspective, existing molecular dynamics approaches suffer from either unaffordable computational costs or insufficient precision. Moreover, inherent limitations such as the considerable random fluctuation of simulated trajectories and the consequential low signal-to-noise ratio pose significant challenges in detecting crucial protein functional motion. Even for the current most advanced Anton 3, under the parallel processing of 512 nodes, for a large system of 1 million atoms, only 100 microseconds of MD simulation can be performed each day[21].

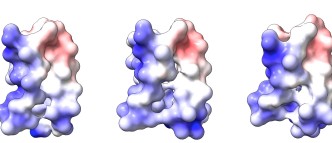

These aforementioned methods give thorough understanding of the natures of proteins by modeling protein molecules as discrete particles. But when studying protein-protein interaction or protein-ligand binding, knowledge about surfaces of proteins is what researchers need. To fulfill such demands, a continuous, high-level representation of molecular shape known as the molecular surface is proposed, as shown in Figure 1.Based on the the concept of "key-lock pairs" emphasizing complementarity of molecular surface[22], the molecular surface models a protein as a continuous shape with geometric and chemical features. One representative method, molecular surface interaction fingerprinting (MaSIF), is a geometric deep learning method that captures fingerprints important for specific biomolecular interactions, enabling accurate and efficient analysis of protein-ligand complexes and potential applications in drug discovery and molecular design[23, 24]. Furthermore, some works investigated the application of end-to-end learning techniques for the analysis of protein surfaces and the identification of potential functional sites, such as drug-target binding sites[25].

Figure 1: The protein surface representation. Showing the protein surface (PDB ID: 1PGA) at different time steps in a molecular dynamics trajectory.

Previously, surface representation was mainly used for studying inter-molecular interactions or assisting molecular docking, and it had not been applied to analysing molecular dynamics simulation data. Inspired by the above works, we propose Dynamical Surface Representation (abbreviated as DSR), a method for modeling protein dynamics by representing protein surfaces with implicit neural networks. Implicit neural representations[26, 27] are a class of deep learning models that can represent complex geometric shapes using a continuous function without requiring an explicit surface representation. It's well-suited for modeling the dynamic shapes of proteins which have complex and varied conformations. Currently, deep learning applications in molecular dynamics (MD) mainly target small molecules, while our method can be applied to very large proteins.

In this paper, we explore the use of implicit neural representations for modeling the dynamic shapes of proteins. To be specific, protein surfaces are represented as a zero-level set of signed distance function (SDF) in DSR. The value of SDF is defined as the minimum distance from a point in Euclidean space to the surface of the object, where the positive value indicates that the point is outside the object, otherwise, it is inside. In addition, we introduce a time variable in DSR model to simulate the change of the protein surface over time, which is the dynamic simulation of the protein. In summary, we use the idea of implicit neural representation and use the neural network as the implicit neural representation of points on the 3D and time region, so as to achieve the purpose of modeling protein dynamic changes in continuous space and time domain.

In summary, our paper's main contribution is two-fold:

- We build an implicit neural network that learns SDF from raw point cloud data in 3D + time domain, which is a both temporally and spatially continuous protein dynamic representation.

- We have achieved dynamic modeling of large proteins by using surface representations, which overcomes the limitations of previous atomic models or coarse-grained models that were restricted to small molecules or proteins.

## 2 DSR

In this section, we present DSR, a dynamical surface representation as implicit neural networks for protein. We first introduce the definition of SDF, and then describe how we use SDF to model proteins. The architecture of the whole method is shown in Figure 2.

### 2.1 SDF Representation

Let $\Omega = [-1, 1]^3$ be a spatial domain, $\tau = [-1, 1]$ a temporal domain, and $\mathcal{M}_t$ be a 3D manifold embedded in $\Omega$ at time $t \in \tau$. For any point $\boldsymbol{x} = (x, y, z) \in \Omega$ at time $t$, the $f_{\mathcal{M}_t} : \Omega \to \mathbb{R}$ is define as

$$f_{\mathcal{M}_t}(\boldsymbol{x}) = \begin{cases} \mathrm{d}(\boldsymbol{x}, \mathcal{M}_t) & \text{if } x \text{ outside } \mathcal{M}_t \\ 0 & \text{if } x \text{ belonging to } \mathcal{M}_t \\ -\mathrm{d}(\boldsymbol{x}, \mathcal{M}_t) & \text{if } x \text{ inside } \mathcal{M}_t \end{cases} \tag{1}$$

where $\mathrm{d}(x, \mathcal{M}_t) := \inf_{y \in \mathcal{M}_t} \mathrm{d}(x, y)$, $\mathrm{d}(x, y) = \|x - y\|_2$.

The zero-level set, which is the surface of the protein at time $t$, is presented by the points where $f_{\mathcal{M}_t}(\cdot) = 0$. That is, the points on the protein surface at time $t$ can be represented as set $\{\boldsymbol{x} \mid f_{\mathcal{M}_t}(\boldsymbol{x}) = 0, \boldsymbol{x} \in \Omega\}$.

### 2.2 Modeling Protein Dynamical Surface with SDFs

The protein surface representation is in the form of a 3D shape. In order to utilize the implicit function form of SDF, a typical approach is to partition the space into a grid, and then compute the SDF value for each grid point. However, for irregular objects, the accurate SDF values cannot be computed and thus approximation algorithms, such as the 8SSEDT algorithm, are usually employed. This method is associated with two limitations: firstly, protein surfaces are often intricate and rugged, and estimation errors can result in larger biases after modeling; secondly, the SDF calculation for dynamic models is linearly dependent on time and cubically dependent on resolution, and pre-computed SDFs will incur high computational costs. Hence, an alternative strategy is used herein, whereby SDF is learned directly from the raw point clouds, rather than utilizing pre-computed SDF for supervised learning. This completely circumvents the need for pre-computing SDF, significantly reduces computational costs, and enhances computational efficiency. In the following, we will elaborate on how to learn SDF directly from the raw point clouds.

For a given input point cloud $\mathcal{X} = \{\boldsymbol{x}_i\}_{i \in I} \subset \mathbb{R}^3$ at time $t$, with point normal vector data (optional), $\mathcal{N} = \{\boldsymbol{n}_i\}_{i \in I} \subset \mathbb{R}^3$, our objective is to find the optimal parameters $\theta$ of an MLP $f(\boldsymbol{x}, t; \theta)$, where $f : \mathbb{R}^{4+d_z} \to \mathbb{R}$, that can accurately estimate a signed distance function to a surface $\mathcal{M}_t$ defined by the point cloud $\mathcal{X}$ and normals $\mathcal{N}$ at time $t$.

The form of our loss function is as follows:

$$\ell(\theta) = \ell_{\mathcal{X}}(\theta) + \lambda \mathbb{E}_{\boldsymbol{x}} \left( \|\nabla_{\boldsymbol{x}} f(\boldsymbol{x}, t; \theta)\| - 1 \right)^2 + \alpha \|z\| \tag{2}$$

where $\lambda > 0$ is a parameter, $\| \cdot \| = \| \cdot \|_2$ is the Euclidean 2-norm, and

$$\ell_{\mathcal{X}}(\theta) = \frac{1}{|I|} \sum_{i \in I} \left( |f(\boldsymbol{x}_i, t; \theta)| + \tau \|\nabla_{\boldsymbol{x}} f(\boldsymbol{x}_i, t; \theta) - \boldsymbol{n}_i\| \right) \tag{3}$$

promotes the vanishing of $f$ on $\mathcal{X}$, and in the presence of normal data (i.e., $\tau = 1$), $\nabla_{\boldsymbol{x}} f$ to the given normals $\mathcal{N}$.

The first term in the summation part of Eq. (3) indicates that the SDF value of the surface points should be as small as possible. The second term represents the loss between the point normal vector and the ground truth.

The second term in Eq. (2) called the *Eikonal term* promotes the gradients $\nabla_{\boldsymbol{x}} f$ to possess a 2-norm of unity, i.e. $\|\nabla_{\boldsymbol{x}} f\| = 1$, which is the gradient property of SDFs. Note that this property is crucial to the loss design, allowing us to avoid using pre-computed SDF values for supervised learning. While its sufficiency has not been fully established in previous literature, we provide a more comprehensive proof in the Appendix B.

The third term $z$ in Eq. (2) is a latent code that we introduce into the model to specify different protein types, and its parameters are learnable. We claim that with sufficient data, we can explore the latent space for greater generalization.

The global minimum of the loss in Eq. (2) will be the solution of the Eikonal partial differential equation, i.e.,

$$\|\nabla_{\boldsymbol{x}} f(\boldsymbol{x})\| = 1, \tag{4}$$

which will also be a signed distance function, where $f$ approaches $0$ on $\mathcal{X}$, with gradients $\mathcal{N}$. The calculation of the expectation is based on some probability distribution $\boldsymbol{x} \sim \mathcal{D}$ in $\mathbb{R}^3$, which will be introduced in Section 3.2.

Throughout the optimization process, we did not use the pre-computed SDF value to supervise the training, which is an end-to-end learning architecture from raw point clouds data to SDF value. In the next section, we will introduce the experiment setup in detail, including the datasets, implementation details, and model evaluation metrics.

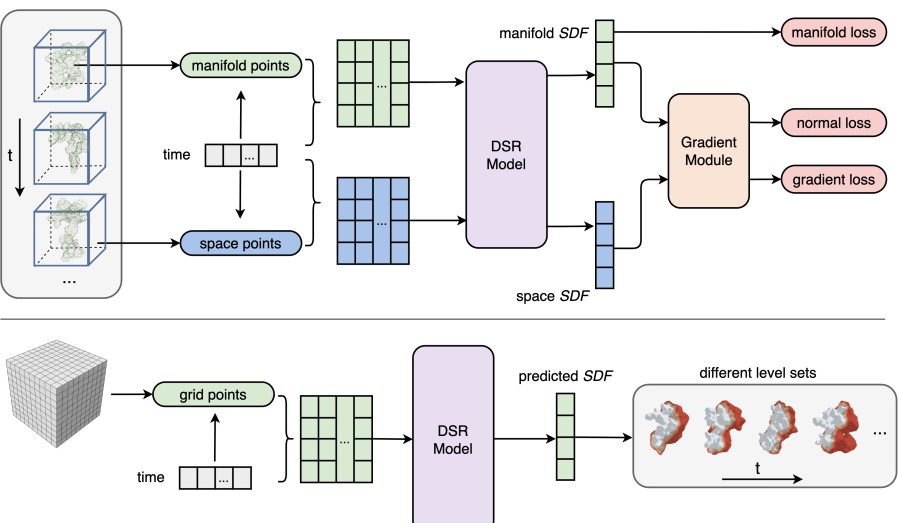

Figure 2: Overview of DSR. The upper part is the training process. First, the manifold and space points are sampled at different times, added the time dimension and be input into the DSR Model to obtain SDF. And then the Gradient Module is used to calculate the derivative of SDF with respect to $(\boldsymbol{x}, t)$. The lower part is the inference process. We first divide the space into a grid based on the desired resolution, then adds the time dimension and inputs them into the DSR Model to predict the SDF. Finally, the Marching Cube algorithm is utilized to generate different level sets, with the zero-level set representing the protein dynamical surface.

## 3 DSR on large-scale proteins simulation data

### 3.1 Dataset.

**500ns Trajectory Data.** This data set[28] currently contains two 100ns atomistic molecular dynamics trajectories of Abeta (40 residues), one wild type and one E22G, following the protocol of the 500ns trajectories published in [29]. The simulation was performed at 310K in a generalized Born implicit solvent. In this work, we used the first 2000 frames of wild type trajectories.

**MDAnalysisData.** MDAnalysisData collects a set of data resources pertaining to computational biophysics, primarily focusing on molecular dynamics (MD) simulations and the structural and dynamic attributes of biomolecules. We used five of these datasets,

(1) AdK equilibrium dataset[30] of 4187 frames,
(2) AdK conformation transition[31] sampled with two methods:
   - Dynamic importance sampling molecular dynamics (DIMS MD) of 102 frames[32],
   - Framework Rigidity Optimized Dynamics Algorithm (FRODA) of 142 frames[33],

(3) A short MD trajectory of I-FABP (intestinal fatty acid binding protein) of 500 frames[34],

(4) A trajectory of the NhaA membrane protein of 5000 frames[35],

(5) Two MD trajectories of YiiP membrane protein, the short one has 900 frames, and the long one has 9000 frames[36]. We opted for the longer one here.

For these 6 trajectories data, we use a fixed size 2000 to crop the frames due to the computational resources, and we keep the full length for those shorter than the cropping size.

**GPCRmd.** The GPCRmd (http://gpcrmd.org/)[37] is an online platform that incorporates web-based visualization capabilities and shares data. The GPCRmd database includes at least one representative structure from each of the four structurally characterized GPCR classes. The GPCRmd platform holds more than 600 GPCR MD simulations from GPCRmd community and individual contributions. Each system was simulated for 500 ns in three replicates (total time 1.5 $\mu s$). We selected one system per GPCR family, and a total of 4 trajectories were used to train the model, of which trajectory ids are 10792_trj_81, 10840_trj_87, 10912_trj_95 and 15711_trj_791 respectively.

**DRYAD_MD.** This dataset contains the trajectory data of 23 proteins simulated by Jumper JM[38]. It is worth noting that the protein trajectories within this dataset exhibit significant oscillations, posing a challenge for model training and resulting in a high failure rate. For this reason, we screened out 14 proteins with mean RMSD[2] less than 1.0 for our experiments.

## 3.2 Implementation Details.

**Problem setup.** In this paper, we use PyMol[39] to obtain the protein surface representation, the points and normal vectors, to train the model. The goal is to obtain a protein surface dynamic model with respect to continuous space and time.

**Architecture.** For representing shapes we used level sets of MLP $f(x, t; \theta) = \mathbb{R}^4 \times \mathbb{R}^{d_z} \to \mathbb{R}$, called DSR Model, with 8 layers, each containing 512 hidden units, and a single skip connection from the input to the middle layer. We set the loss parameters to $\lambda = 0.1, \tau = 1, \alpha = 1e - 3$. And the model architecture is as Figure 2. The activation function between fully connected layers is softplus activation: $x \mapsto \frac{1}{\beta} \ln \left( 1 + e^{\beta x} \right)$, where $\beta = 100$. The initial latent code vector $\mathbf{z}$ of size 192, were sampled from $\mathcal{N}(0, 1.0^2)$. The Gradient Module was implemented by PyTorch Autograd to calculate $\nabla_{\boldsymbol{x}} f(\boldsymbol{x})$.

**Distribution $\mathcal{D}$.** For all experiments, we utilize the distribution $\mathcal{D}$ as defined in the IGR[40] for the expectation in Eq. (2). This distribution is determined by taking the average of a uniform distribution and a sum of Gaussians that are centered at $\mathcal{X}$ with a standard deviation equal to the distance to the $k$-th nearest neighbor (where we set $k = 50$).

**Level set extraction.** We extract the zero (or any other) level set of a trained model $f(x, t; \theta)$ using the *Marching Cubes* algorithm[41] implemented in the *python scikit-image* package, which can use any large-size grid to achieve any high resolution.

## 3.3 Evaluation Metrics.

We use three evaluation metrics commonly used in 3D modeling to evaluate the similarity between two 3D shapes from three aspects: volume, distance, and normal vectors, which are *Volumetric Intersection over Union (IoU)*, *Chamfer distance* and *Normal Consistency (NC)*. These three metrics are all normalized to a range of 0-1, providing a comprehensive evaluation of the model's performance from different perspectives.

**IoU.** IoU compares the reconstructed volume with the ground truth shape (higher is better). For two arbitrary shapes $A, B \subseteq \mathbb{S} \in \mathbb{R}^n$ is attained by:

$$IoU = \frac{|A \cap B|}{|A \cup B|} \tag{5}$$

---

[2]The RMSD here was calculated between consecutive frames to reflect the smoothness of the protein trajectory.

**Chamfer distance.** Chamfer distance is a standard metrics to evaluate the distance between two point sets $\mathcal{X}_1, \mathcal{X}_2 \subset \mathbb{R}^n$ (lower is better).

$$d_C\left(\mathcal{X}_1, \mathcal{X}_2\right) = \frac{1}{2}\left(d_{\overrightarrow{C}}\left(\mathcal{X}_1, \mathcal{X}_2\right) + d_{\overrightarrow{C}}\left(\mathcal{X}_2, \mathcal{X}_1\right)\right) \tag{6}$$

where

$$d_{\overrightarrow{C}}\left(\mathcal{X}_1, \mathcal{X}_2\right) = \frac{1}{|\mathcal{X}_1|} \sum_{\boldsymbol{x}_1 \in \mathcal{X}_1} \min_{\boldsymbol{x}_2 \in \mathcal{X}_2} \|\boldsymbol{x}_1 - \boldsymbol{x}_2\| \tag{7}$$

**NC** NC evaluate estimated surface normals (higher is better). Normal consistency between two normalized unit vectors $n_i$ and $n_j$ is defined as the dot product between the two vectors. For evaluating the surface normals, given the object surface points and normal vectors: $X_{pred} = \{(\boldsymbol{x}_i, \overrightarrow{n_i})\}$, and the ground truth surface points and normal vectors: $X_{gt} = \{(\boldsymbol{y}_j, \overrightarrow{m_j})\}$, the surface normal consistency between $X_{pred}$ and $X_{gt}$, denoted as $\Gamma$, is defined as:

$$\Gamma\left(X_{gt}, X_{\text{pred}}\right) = \frac{1}{|X_{gt}|} \sum_{j \in |X_{gt}|} \left|\overrightarrow{n_j} \cdot \overrightarrow{m}_{\theta\left(\boldsymbol{y}_j, X_{\text{pred}}\right)}\right| \tag{8}$$

where

$$\theta\left(\boldsymbol{y}_j, X_{\text{pred}} := \{(\boldsymbol{x}_i, \overrightarrow{n_i})\}\right) = \arg\min_{i \in |X_{\text{pred}}|} \left\|\boldsymbol{y}_j - \boldsymbol{x}_i\right\|_2^2 \tag{9}$$

## 4 Results and Analysis

We design experiments from three aspects to verify the ability to learn SDF from the raw point clouds, the ability to reconstruct the protein dynamical surface, and the generalization of the model in terms of temporal interpolation and extrapolation. To our knowledge, we are the first to model long-term dynamics of large proteins through surface representation, with no other methods currently available for comparison. Therefore, we conduct the following analysis to verify the learning ability of the DSR model and the effectiveness of protein surface representation. Additionally, to demonstrate that our model is not doing something trivial, we compare our method with linear interpolation on interpolation tasks.

### 4.1 The Ability to Learn SDF

We show the reconstruction results of training on Abeta protein in Figure 3, which demonstrates the ability of our model to learn the signed distance field. The first row in the figure is the reconstruction results of our model, and the second row is the ground truth. It can be seen that our model has the ability to reconstruct the dynamic shape of the protein at different moments, and can restore a clearer and more delicate shape than the ground truth. Furthermore, we show some videos of this protein on the webpage and discuss the effect of normal vectors on learning process in the Appendix D.

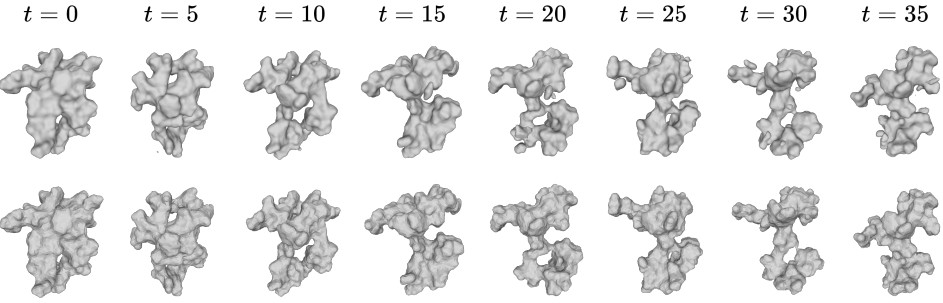

Figure 3: Given different times as input, the zero-level set visualization of the SDF value predicted by the model for protein abeta. The first row is the reconstruction result of our model, and the second row is the ground truth.

Note that SDF represents the shortest distance (signed) from the surface of the object, which means that different level sets form surfaces at certain distances from the manifold surface. As shown

in Figure 4 (a), the visualization of different level sets of SDF values predicted by our model is consistent with the meaning expressed by SDF. In addition, our model is continuous with respect to spatial and can use to reconstruct and predict for arbitrary space resolutions. Figure 4(b) shows the reconstruction of a protein at different resolutions at a certain time. We also calculated three evaluation metrics, namely IoU, Chamfer distance and NC, which have values of 0.9361±0.0265, 0.0009±0.0008 and 0.9967±0.0011 respectively.

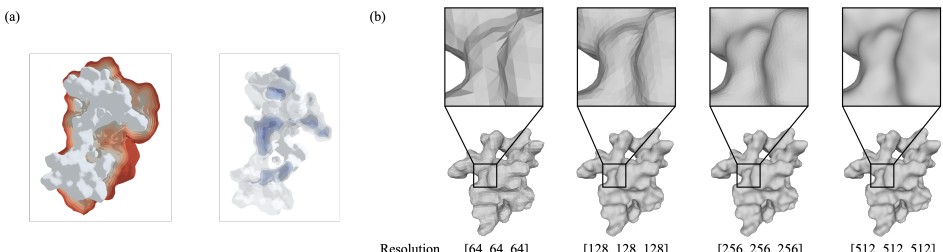

Figure 4: (a) The visualization of different level-set of the SDF predicted by our model; positive level sets are in red; negative ones are in blue; the zero level set, represents the approximated surface in white. (b) The reconstruction of a protein at different resolutions.

## 4.2 Reconstruction across Time

In this experiment, we validate the ability of our trained model to reconstruct the protein shapes in training frames (see Table 1). For data AdK equilibrium, NhaA, YiiP, and DRYAD_MD, we used the first 1500 frames of each trajectory and diluted them by ten times. That is, selecting 150 frames for each trajectory for training. Similarly, every second frame was selected within the first 80 frames for DIMS, every third frame was chosen within the first 120 frames for FRODA, and every fifth frame was selected within the first 400 frames for I-FABP.

As we can see from the Table 1, the model can reconstruct well in many cases. For example, DIMS, the IoU value is as high as 0.9316, which means that the shape reconstructed by the model is almost completely consistent with the ground truth on the entire timeline.

Table 1: Evaluation on the surface reconstruction across time.

| | IoU↑ | Chamfer_dist↓ | NC↑ | | IoU↑ | Chamfer_dist↓ | NC↑ |
|---|---|---|---|---|---|---|---|
| DIMS | 0.9316±0.0070 | 0.0003±0.0001 | 0.9671±0.0050 | I-FABP | 0.8907±0.0071 | 0.0005±0.0001 | 0.9382±0.0072 |
| YiiP | 0.8425±0.0161 | 0.0003±0.0001 | 0.8737±0.0164 | FRODA | 0.8318±0.0286 | 0.0004±0.0001 | 0.9098±0.0175 |
| NahA | 0.8265±0.0202 | 0.0005±0.0001 | 0.8372±0.0245 | adk_equi | 0.7526±0.0396 | 0.0020±0.0009 | 0.7914±0.0308 |
| protG | 0.6580±0.0695 | 0.0036±0.0017 | 0.8287±0.0595 | ntl9 | 0.6342±0.1244 | 0.0066±0.0051 | 0.8165±0.0911 |
| bba | 0.6163±0.1481 | 0.0057±0.0073 | 0.8158±0.1044 | cspa | 0.6136±0.0682 | 0.0074±0.0036 | 0.8047±0.0527 |
| T0765 | 0.5997±0.0923 | 0.0039±0.0022 | 0.7991±0.0757 | ww | 0.5920±0.1056 | 0.0088±0.0055 | 0.8196±0.0691 |
| T0855 | 0.5585±0.0898 | 0.0117±0.0095 | 0.7804±0.0647 | T0773 | 0.5289±0.0833 | 0.0123±0.0091 | 0.7404±0.0632 |
| T0816 | 0.5284±0.1366 | 0.0083±0.0071 | 0.7368±0.0979 | gpW | 0.4836±0.1253 | 0.0093±0.0072 | 0.7176±0.0905 |
| bbl | 0.4603±0.1422 | 0.0094±0.0067 | 0.6968±0.0896 | hyp | 0.4578±0.1154 | 0.0099±0.0066 | 0.7104±0.0873 |
| T0769 | 0.4535±0.0871 | 0.0191±0.0101 | 0.7102±0.0738 | T0771 | 0.4219±0.0711 | 0.0111±0.0073 | 0.6537±0.0491 |

We analyze the effect of RMSD on model training quality, indicating that data containing high levels of noise and significant vibration pose challenges to model training. As illustrated in the Figure 5, the model shows superior performance on smoother and less noisy trajectory data.

## 4.3 Generalization Ability

Our model has generative capabilities to predict missing or future trajectories of a protein. Here we obtain different samples in two aspects, interpolation and extrapolation of time.

### 4.3.1 Interpolation

Our model enable the prediction of protein shape at any given time due to its temporal continuity. This is very useful when we only know the structures of two moments but not what the intermediate process is. During training sample selection, we extract one frame at intervals of several frames to evaluate the model's interpolation capability in the temporal space. To demonstrate that our model

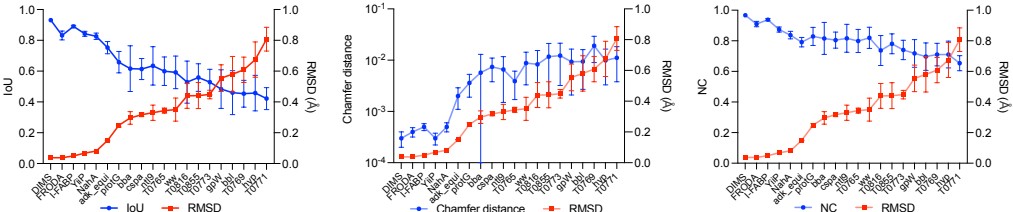

Figure 5: The correlation between RMSD and three metrics was evaluated across various protein trajectories, revealing a significant relationship between the metrics and RMSD, highlighting that smaller RMSD and system noise levels correspond to improved model performance and easier modeling of protein surface dynamics.

learns the protein's inherent dynamic variations and is not just performing trivial tasks, we compared it with linear interpolation, as shown in Figure 6, even though there were no previous works for direct comparison. The evaluation results of our model and linear interpolation are shown in Appendix Table 4, 5 respectively, and the performance of interpolation has reached a level comparable to that of reconstruction.

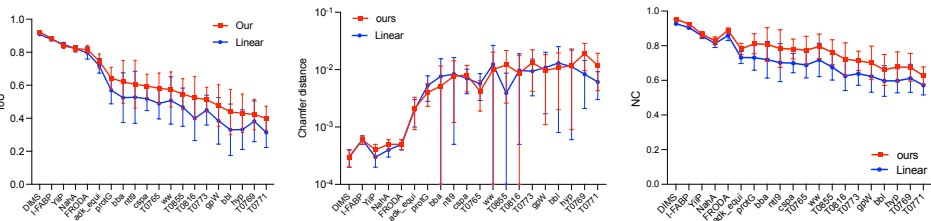

Figure 6: The three evaluation metrics for the interpolation task are shown in the plot, with the red line representing our method and the blue line representing linear interpolation.

### 4.3.2 Future Prediction

Our model can not only interpolate missing frames in the middle, but also predict future dynamics by extrapolating over time. To validate this, we use models trained on the MDAnalysisData and DRYAD_MD datasets to predict future frames. Table 2 shows the prediction results of the model for the next 10 frames and the next $11 \sim 20$ frames on some proteins. Due to space limitations of the main text, the full results are presented in the Appendix E.2.

Table 2: Evaluation on the future 10 frames and $11 \sim 20$ frames

|  | Future 10 frames | | | Future 11~20 | | |
|---|---|---|---|---|---|---|
|  | IoU↑ | Chamfer_dist↓ | NC↑ | IoU↑ | Chamfer_dist↓ | NC↑ |
| DIMS | 0.8467±0.0423 | 0.0010±0.0005 | 0.8666±0.0504 | 0.6835±0.0551 | 0.0055±0.0020 | 0.6932±0.0474 |
| I-FABP | 0.8878±0.0142 | 0.0006±0.0001 | 0.9271±0.0159 | 0.8444±0.0153 | 0.0012±0.0002 | 0.8797±0.0164 |
| FRODA | 0.7660±0.0502 | 0.0012±0.0006 | 0.8267±0.0490 | 0.6043±0.0474 | 0.0044±0.0012 | 0.6783±0.0384 |

### 4.3.3 A Case Study on GPCRmd

Furthermore, we have explored the model's simultaneous simulation of multiple protein trajectories. Here we use the four protein trajectories data in GPCRmd dataset. The model trains different latent vectors to represent different proteins. In addition, in selecting training frames, we have employed another strategy, wherein the interval between the frames utilized for training is gradually increased with the sequence.

As shown in Figure 7, our model can learn the shapes of multiple proteins simultaneously, and it can recover the fine details of the protein surfaces well despite their ruggedness. An interesting finding is that the trajectories obtained from molecular dynamics simulations often have a lot of random jittering or noise, while the trajectories interpolated by our model can well fit the main functional

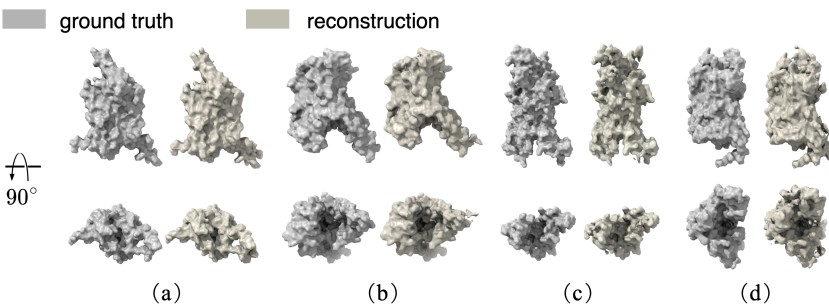

Figure 7: Comparison of ground truth and reconstruction for GPCRmd data at $t = 2000$. From (a) to (d) are 10792_trj_81, 10912_trj_95, 15711_trj_791 and 16105_trj_848 respectively, ground truth and reconstruction (top row) after rotating 90 degrees (bottom row) clockwise along the horizontal axis.

motion of the protein without noise. We put the detailed results and discussion in the Appendix E.3, video demonstration on the webpage.

## 5 Discussions

**Molecular surface Representation**  The molecular surface representation is commonly adopted for tasks involving molecular interfaces[42], where non-covalent interactions (e.g., hydrophobic interactions) play a decisive role[43]. Non-Euclidean convolutional neural networks[44] and point cloud-based learning models[45] have been applied to encode the molecular surface for downstream applications, e.g., protein binding site prediction[46]. However, it has not been applied to molecular dynamic simulation, especially protein. In fact, this work explored the more essential application scenarios of molecular surface representation, that is, we believe that chemical molecules are more suitable to be regarded as electron clouds that can be represented by 3D shapes.

**Deep learning molecular dynamics**  Deep learning is being used more in molecular dynamics simulations, which has shown promising results. There have been several approaches, such as Behler-Parrinello network[47], DTNN[48], and SchNet[49], that focus on predicting molecular properties and potential energy. However, these methods are limited to small molecules and may not work for proteins, which have many atoms. To address this issue, several coarse-grained models have been proposed, such as CGnet[17], DeePCG[50], ARCG[51], and a CG auto-encoder[52]. Recent works, flow-matching[53] and DFF[54], have used normalizing flows and diffusion models to model coarse-grained force fields for dynamic simulation of small proteins. Another approach, DiffMD[55], can predict simulation trajectories without energy or forces. However, these methods are currently limited to simulating small proteins due to computational constraints and are not yet practical for real-world applications, while the method in this paper provides a new idea for long-term modeling of large proteins.

**Implicit neural representations**  SIREN[56] and NeRF[26] have led to the popularity of implicit neural representations (INR) for tasks like new view synthesis and geometry reconstruction. INR involves creating a continuous function that can map coordinates to high-frequency information. NeRF's highly expressive nature has allowed it to be effectively employed in numerous fields, such as image processing[57] (e.g., compression, denoising, super-resolution, inpainting), video processing[58], medical imaging[59], etc. Signed distance function (SDF) is a type of INR that characterizes an object's shape well. DeepSDF[60] uses INR to model SDF, and IGR[40] simplifies the modeling of SDF by using its gradient property. Drawing inspiration from the aforementioned studies, our work learns a unified implicit representation of proteins that will facilitate future large-scale modeling of protein dynamics like NeRF's contribution in vision.

## 6 Conclusions and Future work

We introduced a method for learning implicit neural representations of protein surface dynamics in continuous spatial and temporal space. Experimental results show that our method is very effective in reconstructing proteins with smooth dynamic changes, and has a certain ability to interpolate in time.

In addition, we found that our model tends to model major conformation changes of proteins rather than those noise-like random vibrations. The case study of GPCRmd shows that the model has the potential to learn the dynamics of a class of similar proteins, such as a protein family.

The main limitation of this method is that it is sensitive to noise in protein dynamic trajectories, as discussed in Section 4.2. In addition, the generalization of the model needs to be further improved. The first is the temporal generalization, especially future prediction. So far, there is no effective method that can better predict the future in various motion prediction tasks. We believe that modeling motion or velocity fields can help future predictions for a longer time.

We anticipate DSR as a powerful tool for building general-purpose protein surface models, and hope that our work helps shed some light on a more efficient and generalizable protein representation.

## Acknowledgments

This work was supported by Mathematical Intelligence Application Laboratory, MIALAB, Institute for Mathematical Sciences, Renmin University of China, Beijing Academy of Artificial Intelligence and Public Computing Cloud of Renmin University of China.

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
