# Appendix

## A  Motivations

The traditional methods for molecular dynamics simulation, such as solving the Schrödinger equation, the Born-Oppenheimer approximation method, and Density Functional Theory (DFT), are computationally expensive and time-consuming. To provide a more detailed explanation, we can give specific numbers to illustrate the computational complexity of these methods. For instance, solving the Schrödinger equation directly for a system of 100 atoms would require solving a matrix with approximately $1.26 \times 10^6$ elements, which is computationally intractable even with supercomputers. The Born-Oppenheimer approximation, which separates electronic and nuclear motion, is also computationally expensive. For example, a single energy minimization calculation using this approximation for a protein containing 10,000 atoms can take several hours on a high-performance computing cluster. DFT, which is a widely used approximation for solving the Schrödinger equation, has a time complexity of $O(n^3)$, where $n$ is the number of atoms in the system. For example, a DFT calculation for a system of 1000 atoms would require approximately 1 billion calculations.

Since traditional biochemists tend to focus on atomic representations of proteins, the computational complexity is closely related to the number of atoms in the system. However, we have converted the representation to a continuous function of the protein surface, which can significantly reduce the computational complexity.

## B  Theory Analysis

**Theorem 1.** *Given a manifold $\Omega$, the implicit signed distance function is defined as $\phi(\boldsymbol{x})$, $x \in \mathbb{R}^n$ with $\phi(\boldsymbol{x}) < 0$ in the interior regin $\Omega^-$, $\phi(\boldsymbol{x}) > 0$ in the exterior regin $\Omega^+$, and $\phi(\boldsymbol{x}) = 0$ on the boundary $\partial\Omega$, then $|\nabla\phi(\boldsymbol{x})| = 1$.*

$$\phi(\boldsymbol{x}) = \begin{cases} d(\boldsymbol{x}, \Omega) & \text{if } \boldsymbol{x} \in \Omega^+ \\ 0 & \text{if } \boldsymbol{x} \in \Omega \\ -d(\boldsymbol{x}, \Omega) & \text{if } x \in \Omega^- \end{cases} \tag{10}$$

*where $d(x, \Omega) := \min(\|\boldsymbol{x} - \boldsymbol{x}_I\|)$ for all $\boldsymbol{x}_I \in \partial\Omega$.*

*Proof.* Firstly, for a given point $\boldsymbol{x}$, suppose that $\boldsymbol{x}_c$ is the only point on the surface closest to $\boldsymbol{x}$. Then for every point $\boldsymbol{y}$ on the line segment between points $\boldsymbol{x}$ and $\boldsymbol{x}_c$, the point on the surface closest to $\boldsymbol{y}$ is still $\boldsymbol{x}_c$.

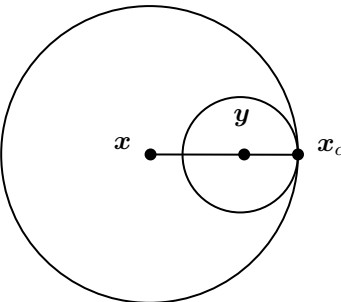

Figure 8: $\boldsymbol{x}_C$ is the closest surface point to $\boldsymbol{x}$ and $\boldsymbol{y}$.

As shown in Figure 8, inside a sphere with $\boldsymbol{x}\boldsymbol{x}_c$ as radius, there will be no point on the surface closer to $\boldsymbol{x}$. Similarly, within the sphere with radius $\boldsymbol{y}\boldsymbol{x}_c$, there will not be any point on the surface that is closer to point $\boldsymbol{y}$. Otherwise, the distance between the point and $\boldsymbol{x}$ would be less than the distance between $\boldsymbol{x}\boldsymbol{x}_c$, which contradicts the assumption. Therefore, it can be seen that the line segment $\boldsymbol{x}\boldsymbol{x}_c$ is the shortest path from point $\boldsymbol{x}$ to the surface, and it is also the direction in which $\phi(\boldsymbol{x})$ changes the fastest (i.e., for $x \in \Omega^-$, $\phi(\boldsymbol{x})$ increases the fastest, and for $x \in \Omega^+$, $\phi(\boldsymbol{x})$ decreases the fastest).

Then,

$$|\nabla\phi(\boldsymbol{x})| = \left| \frac{\partial\phi(\boldsymbol{x})}{\partial\left(\overrightarrow{\boldsymbol{x}\boldsymbol{x}_{\boldsymbol{c}}}\right)} \right|$$

$$= \left| \lim_{\substack{\boldsymbol{y}\to\boldsymbol{x}\\ \boldsymbol{y}\in\boldsymbol{x}\boldsymbol{x}_c}} \frac{\phi(\boldsymbol{y})-\phi(\boldsymbol{x})}{d(\boldsymbol{y},\boldsymbol{x})} \right|$$

$$= \left| \lim_{\substack{\boldsymbol{y}\to\boldsymbol{x}\\ \boldsymbol{y}\in\boldsymbol{x}\boldsymbol{x}_c}} \frac{d\left(\boldsymbol{y},\boldsymbol{x_c}\right)-d\left(\boldsymbol{x},\boldsymbol{x_c}\right)}{d(\boldsymbol{y},\boldsymbol{x})} \right| \qquad (11)$$

$$= \left| \lim_{\substack{\boldsymbol{y}\to\boldsymbol{x}\\ \boldsymbol{y}\in\boldsymbol{x}\boldsymbol{x}_c}} \frac{d(\boldsymbol{y},\boldsymbol{x})}{d(\boldsymbol{y},\boldsymbol{x})} \right| = 1$$

Secondly, if there are more than one point on the surface that is closest to x, let these points be $\{\boldsymbol{x}_{c1}, \boldsymbol{x}_{c2}, \cdots, \boldsymbol{x}_{cN}\}$. Similarly, within the sphere with radius $\boldsymbol{x}\boldsymbol{x}_{ci}, i \in 1, 2, \cdots, N$, there will not be any point on the surface closer to $\boldsymbol{x}$ than $\{\boldsymbol{x}_{c1}, \boldsymbol{x}_{c2}, \cdots, \boldsymbol{x}_{cN}\}$. Therefore, gradient direction should be the direction with the maximum numerical value among the directional derivatives in $\{\boldsymbol{x}\boldsymbol{x}_{c1}, \boldsymbol{x}\boldsymbol{x}_{c2}, \cdots, \boldsymbol{x}\boldsymbol{x}_{cN}\}$, i.e.

$$|\nabla\phi(\boldsymbol{x})| = \max_i \left| \frac{\partial\phi(\boldsymbol{x})}{\partial\left(\overrightarrow{\boldsymbol{x}\boldsymbol{x_{ci}}}\right)} \right| = \max_i \left| \lim_{\substack{\boldsymbol{y}\to\boldsymbol{x}\\ \boldsymbol{y}\in\boldsymbol{x}\boldsymbol{x}_{ci}}} \frac{\phi(\boldsymbol{y})-\phi(\boldsymbol{x})}{d(\boldsymbol{y},\boldsymbol{x})} \right| = 1 \qquad (12)$$

$\square$

## C  Experiment Details

For each protein surface representation trajectory used for training, we scale the space to $[-1,1]^3$ and the time to $[-1,1]$. When reconstructing, interpolating, and extrapolating, for time, we scale at the same scale as training, and for space, we divide the $[-1,1]^3$ space into grid points according to a given resolution. We use the Marching Cube algorithm to extract the zero-level set of the SDF value output by the model, and scale it back to the original space size.

The training was done on a single Nvidia-V100 GPU with PyTorch deep learning framework[61]. For training, we use the Adam optimizer[62], where initial learning rate of sdf training is 5e-3 with a decay of 0.5 every 200 epochs and initial learning rate of latent code is 1e-3 with a decay of 0.5 every 200 epochs. We trained each model for 4000 epochs. Because our model is continuous in time and space, we can reconstruct or interpolate or extrapolate protein shapes at any time and at any resolution.

For experiments of GPCRmd, we use different latent code to specify different sequences. When doing inference, you can use different latent codes to perform tasks corresponding to the sequence by specifying the sequence id.

Table 3: The architecture of our neural network.

| Layer | Input shape | Output shape | |
|---|---|---|---|
| Dense layer | 196 | 512 | $(x,t) \in \mathbb{R}^4$, latent code $\in \mathbb{R}^{192}$ |
| Dense layer | 512 | 512 | |
| Dense layer | 512 | 512 | |
| Dense layer | 512 | 316 | skip connection from the first layer |
| Dense layer | 512 | 512 | |
| Dense layer | 512 | 512 | |
| Dense layer | 512 | 512 | |
| Dense layer | 512 | 1 | |
| Activation | | Softplus(beta=100, threshold=20) | |

# D   Learning Ability

In order to illustrate that our method has the ability to learn the shape of objects, we visualize the training process in Figure9. As can be seen from the Figure9, with the training process, the model can reconstruct the shape of the protein in more detail. In addition, using normal vectors as constraints in loss function during training will make the model achieve better results faster.

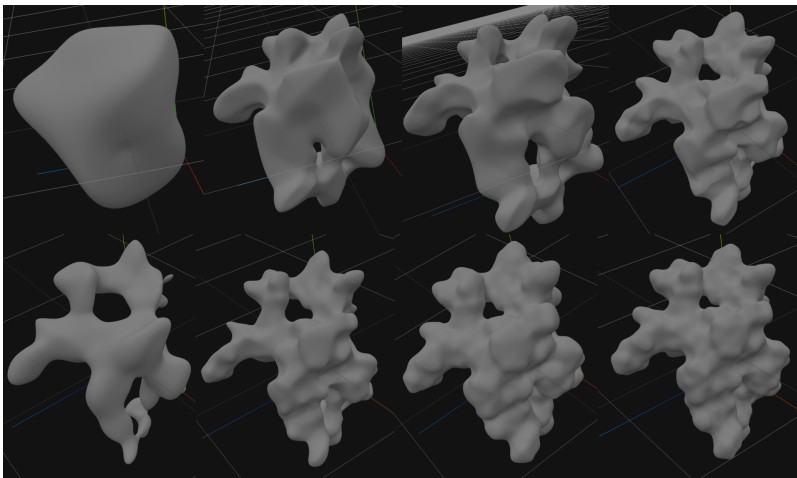

Figure 9: The shape visualization of Abeta. First row: the result of training epoch 100, 500, 1000 and 2000 epochs respectively without normal vector; second row: the result of training epoch 100, 300, 500 and 1000 epochs respectively with normal vector.

# E Detailed results

## E.1 Interpolation

Table 4: Evaluation on the surface interpolation between trained frames by our model.

|        | IoU↑           | Chamfer_dist↓   | NC↑            |          | IoU↑           | Chamfer_dist↓   | NC↑            |
|--------|----------------|-----------------|----------------|----------|----------------|-----------------|----------------|
| DIMS   | 0.9225±0.0035  | 0.0003±0.0001   | 0.9527±0.0034  | I-FABP   | 0.8833±0.0078  | 0.0006±0.0001   | 0.9254±0.0066  |
| YiiP   | 0.8387±0.0157  | 0.0004±0.0001   | 0.8659±0.0157  | FRODA    | 0.8159±0.0234  | 0.0005±0.0001   | 0.8885±0.0135  |
| NahA   | 0.8220±0.0191  | 0.0005±0.0001   | 0.8287±0.0219  | adk_equi | 0.7459±0.0436  | 0.0021±0.0012   | 0.7816±0.0327  |
| protG  | 0.6434±0.0655  | 0.0040±0.0017   | 0.8113±0.0580  | ntl9     | 0.6059±0.1452  | 0.0077±0.0065   | 0.7838±0.1083  |
| bba    | 0.6204±0.1296  | 0.0052±0.0064   | 0.8074±0.0972  | cspa     | 0.5957±0.0714  | 0.0080±0.0039   | 0.7812±0.0574  |
| T0765  | 0.5821±0.0921  | 0.0042±0.0023   | 0.7747±0.0797  | ww       | 0.5744±0.1061  | 0.0100±0.0102   | 0.7955±0.0720  |
| T0855  | 0.5458±0.0943  | 0.0121±0.0095   | 0.7629±0.0716  | T0773    | 0.5127±0.0755  | 0.0132±0.0090   | 0.7130±0.0559  |
| T0816  | 0.5245±0.1289  | 0.0087±0.0088   | 0.7221±0.0951  | gpW      | 0.4769±0.1262  | 0.0097±0.0086   | 0.7008±0.0967  |
| bbl    | 0.4380±0.1372  | 0.0108±0.0088   | 0.6605±0.0895  | hyp      | 0.4304±0.1163  | 0.0119±0.0110   | 0.6776±0.0865  |
| T0769  | 0.4229±0.0932  | 0.0191±0.0095   | 0.6752±0.0801  | T0771    | 0.4006±0.0729  | 0.0118±0.0075   | 0.6271±0.0503  |

Table 5: Evaluation on the surface interpolation between trained frames by linear interpolation.

|        | IoU↑           | Chamfer_dist↓   | NC↑            |          | IoU↑           | Chamfer_dist↓   | NC↑            |
|--------|----------------|-----------------|----------------|----------|----------------|-----------------|----------------|
| DIMS   | 0.9084±0.0071  | 0.0003±0.0001   | 0.9266±0.0079  | I-FABP   | 0.8770±0.0076  | 0.0006±0.0001   | 0.9053±0.0082  |
| YiiP   | 0.8475±0.0126  | 0.0003±0.0001   | 0.8560±0.0143  | FRODA    | 0.7961±0.0368  | 0.0005±0.0001   | 0.8593±0.0292  |
| NahA   | 0.8219±0.0192  | 0.0004±0.0001   | 0.8129±0.0237  | adk_equi | 0.7181±0.0440  | 0.0020±0.0010   | 0.7320±0.0352  |
| protG  | 0.5694±0.0820  | 0.0053±0.0025   | 0.7309±0.0719  | ntl9     | 0.5273±0.1578  | 0.0084±0.0079   | 0.7021±0.1100  |
| bba    | 0.5249±0.1506  | 0.0076±0.0079   | 0.7193±0.1056  | cspa     | 0.5189±0.0728  | 0.0070±0.0032   | 0.6994±0.0556  |
| T0765  | 0.4899±0.0937  | 0.0057±0.0028   | 0.6885±0.0745  | ww       | 0.5079±0.1433  | 0.0124±0.0142   | 0.7177±0.0965  |
| T0855  | 0.4674±0.1156  | 0.0039±0.0048   | 0.6762±0.0746  | T0773    | 0.4496±0.0904  | 0.0094±0.0059   | 0.6389±0.0672  |
| T0816  | 0.3994±0.1344  | 0.0097±0.0092   | 0.6259±0.0845  | gpW      | 0.3846±0.1411  | 0.0109±0.0092   | 0.6228±0.0912  |
| bbl    | 0.3304±0.1558  | 0.0130±0.0122   | 0.5964±0.0881  | hyp      | 0.3313±0.1201  | 0.0113±0.0107   | 0.5974±0.0691  |
| T0769  | 0.3820±0.1236  | 0.0083±0.0062   | 0.6122±0.0808  | T0771    | 0.3147±0.0914  | 0.0061±0.0031   | 0.5723±0.0564  |

## E.2 Future Prediction

In this section we list the three evaluation metrics for the prediction results of all other proteins in the next 100 frames. Tables 6, 7, and 8 show the results of IoU, chamfer distance, and NC respectively. As shown in the table, the results decrease slightly as time goes into the future, which is not surprising. It is worth noting that even when predicting the next 100 frames of some proteins, the prediction results do not drop much, such as the Yiip, NahA and adk_equi, which also shows that our model has certain expression power and stability.

Table 6: The future prediction results on IoU

| Future frames | 10 | 11∼20 | 21∼30 | 31∼40 | 41∼50 |
|---|---|---|---|---|---|
| YiiP | 0.8482±0.0041 | 0.8293±0.0067 | 0.8068±0.0087 | 0.8128±0.0091 | 0.8122±0.0090 |
| NahA | 0.7922±0.0137 | 0.7914±0.0133 | 0.7640±0.0141 | 0.7589±0.0071 | 0.7465±0.0092 |
| adk_equi | 0.7417±0.0289 | 0.7304±0.0285 | 0.6582±0.0680 | 0.7107±0.0385 | 0.7080±0.0589 |
| protG | 0.6666±0.0579 | 0.6366±0.0641 | 0.6390±0.0371 | 0.6257±0.0611 | 0.6203±0.0614 |
| ntl9 | 0.5829±0.2385 | 0.6712±0.1451 | 0.6997±0.0593 | 0.6069±0.1673 | 0.6092±0.1806 |
| bba | 0.4958±0.0546 | 0.5088±0.0432 | 0.5334±0.0261 | 0.4973±0.0315 | 0.4628±0.1062 |
| cspa | 0.5467±0.0935 | 0.5237±0.0624 | 0.4914±0.0479 | 0.5332±0.0494 | 0.4872±0.0744 |
| T0765 | 0.4820±0.0941 | 0.4638±0.0877 | 0.4684±0.0628 | 0.4544±0.0476 | 0.4241±0.0442 |
| ww | 0.3716±0.1133 | 0.3163±0.0826 | 0.3176±0.0620 | 0.2753±0.0162 | 0.2729±0.0234 |
| T0855 | 0.4215±0.0368 | 0.3631±0.0374 | 0.3221±0.0286 | 0.2899±0.0308 | 0.2355±0.0398 |
| T0773 | 0.4364±0.0982 | 0.4011±0.0848 | 0.4272±0.0880 | 0.4169±0.0839 | 0.3802±0.0727 |
| T0816 | 0.5301±0.0805 | 0.5502±0.0557 | 0.5187±0.0574 | 0.4652±0.1149 | 0.4366±0.1391 |
| gpW | 0.4604±0.1025 | 0.4644±0.1483 | 0.4785±0.1504 | 0.5072±0.0857 | 0.4492±0.0957 |
| bbl | 0.2557±0.2497 | 0.3817±0.2172 | 0.4064±0.2026 | 0.1792±0.0579 | 0.1916±0.0453 |
| hyp | 0.3189±0.0746 | 0.3185±0.0751 | 0.2889±0.0693 | 0.2498±0.0490 | 0.2650±0.0370 |
| T0769 | 0.4256±0.0706 | 0.4216±0.0867 | 0.3700±0.0666 | 0.4157±0.0918 | 0.4313±0.0668 |
| T0771 | 0.4031±0.0798 | 0.3738±0.0724 | 0.3543±0.0717 | 0.3596±0.0668 | 0.3538±0.0569 |
| Future frames | 51∼60 | 61∼70 | 71∼80 | 81∼90 | 91∼100 |
| YiiP | 0.7986±0.0071 | 0.7744±0.0116 | 0.7686±0.0058 | 0.7698±0.0076 | 0.7546±0.0066 |
| NahA | 0.7637±0.0079 | 0.7551±0.0046 | 0.7402±0.0081 | 0.7438±0.0072 | 0.7330±0.0097 |
| adk_equi | 0.7625±0.0153 | 0.7814±0.0177 | 0.7635±0.0170 | 0.7402±0.0273 | 0.7144±0.0343 |
| protG | 0.6083±0.0342 | 0.6159±0.0276 | 0.6216±0.0477 | 0.6165±0.0466 | 0.5892±0.0644 |
| ntl9 | 0.5884±0.1663 | 0.5821±0.1629 | 0.5614±0.1328 | 0.6173±0.1268 | 0.6306±0.0274 |
| bba | 0.4950±0.0140 | 0.4606±0.0734 | 0.4469±0.0495 | 0.4138±0.0639 | 0.4439±0.0159 |
| cspa | 0.4631±0.0683 | 0.4704±0.0308 | 0.4312±0.0597 | 0.4157±0.0689 | 0.3389±0.0467 |
| T0765 | 0.4523±0.0361 | 0.4199±0.0318 | 0.4179±0.0315 | 0.3732±0.0292 | 0.3463±0.0396 |
| ww | 0.2549±0.0273 | 0.2364±0.0279 | 0.2225±0.0154 | 0.2175±0.0294 | 0.2076±0.0366 |
| T0855 | 0.2401±0.0407 | 0.1940±0.0303 | 0.1956±0.0155 | 0.1609±0.0178 | 0.1460±0.0244 |
| T0773 | 0.4414±0.0686 | 0.3944±0.0777 | 0.3706±0.0788 | 0.3455±0.0454 | 0.3166±0.0478 |
| T0816 | 0.4738±0.0916 | 0.4706±0.1068 | 0.4301±0.1223 | 0.4740±0.0240 | 0.4167±0.0849 |
| gpW | 0.3396±0.1590 | 0.4465±0.1135 | 0.4157±0.0684 | 0.4349±0.1060 | 0.4121±0.0894 |
| bbl | 0.2384±0.0626 | 0.2008±0.0631 | 0.1802±0.0676 | 0.1861±0.0450 | 0.1819±0.0530 |
| hyp | 0.2171±0.0733 | 0.2445±0.0203 | 0.2240±0.0347 | 0.2035±0.0409 | 0.1915±0.0321 |
| T0769 | 0.4022±0.0699 | 0.3642±0.0584 | 0.3614±0.0507 | 0.3194±0.0527 | 0.3391±0.0438 |
| T0771 | 0.3373±0.0570 | 0.3053±0.0540 | 0.3068±0.0359 | 0.2831±0.0325 | 0.2603±0.0353 |

Table 7: The future prediction results on Chamfer distance

| Future frames | 10 | 11~20 | 21~30 | 31~40 | 41~50 |
|---|---|---|---|---|---|
| YiiP | 0.0003±2.1863 | 0.0004±3.7481 | 0.0005±5.4920 | 0.0005±5.3535 | 0.0005±4.2672 |
| NahA | 0.0007±7.7686 | 0.0007±8.1652 | 0.0009±8.1229 | 0.0010±4.5848 | 0.0011±8.0088 |
| adk_equi | 0.0021±0.0006 | 0.0026±0.0006 | 0.0050±0.0027 | 0.0029±0.0009 | 0.0031±0.0014 |
| protG | 0.0036±0.0017 | 0.0042±0.0017 | 0.0041±0.0014 | 0.0046±0.0018 | 0.0045±0.0017 |
| ntl9 | 0.0112±0.0150 | 0.0044±0.0066 | 0.0023±0.0009 | 0.0069±0.0086 | 0.0065±0.0082 |
| bba | 0.0088±0.0026 | 0.0081±0.0016 | 0.0073±0.0009 | 0.0087±0.0009 | 0.0126±0.0105 |
| cspa | 0.0130±0.0028 | 0.0140±0.0022 | 0.0152±0.0019 | 0.0134±0.0014 | 0.0161±0.0044 |
| T0765 | 0.0071±0.0035 | 0.0078±0.0029 | 0.0075±0.0025 | 0.0078±0.0019 | 0.0083±0.0020 |
| ww | 0.0170±0.0028 | 0.0196±0.0037 | 0.0201±0.0021 | 0.0202±0.0019 | 0.0209±0.0011 |
| T0855 | 0.0131±0.0012 | 0.0142±0.0015 | 0.0148±0.0023 | 0.0159±0.0019 | 0.0186±0.0031 |
| T0773 | 0.0159±0.0068 | 0.0179±0.0078 | 0.0177±0.0106 | 0.0162±0.0080 | 0.0160±0.0057 |
| T0816 | 0.0286±0.0043 | 0.0310±0.0044 | 0.0318±0.0044 | 0.0345±0.0041 | 0.0340±0.0066 |
| gpW | 0.0104±0.0021 | 0.0131±0.0037 | 0.0117±0.0024 | 0.0122±0.0033 | 0.0140±0.0036 |
| bbl | 0.0276±0.0203 | 0.0213±0.0265 | 0.0155±0.0223 | 0.0283±0.0222 | 0.0209±0.0057 |
| hyp | 0.0354±0.0025 | 0.0378±0.0090 | 0.0387±0.0104 | 0.0372±0.0050 | 0.0371±0.0050 |
| T0769 | 0.0142±0.0028 | 0.0123±0.0020 | 0.0104±0.0024 | 0.0093±0.0019 | 0.0079±0.0011 |
| T0771 | 0.0083±0.0025 | 0.0087±0.0023 | 0.0079±0.0011 | 0.0090±0.0012 | 0.0089±0.0010 |
| Future frames | 51~60 | 61~70 | 71~80 | 81~90 | 91~100 |
| YiiP | 0.0006±4.0775 | 0.0007±7.4013 | 0.0008±3.4454 | 0.0008±6.6268 | 0.0009±5.4942 |
| NahA | 0.0010±6.2155 | 0.0011±4.4547 | 0.0013±7.3425 | 0.0013±5.2765 | 0.0014±6.8101 |
| adk_equi | 0.0017±0.0002 | 0.0015±0.0002 | 0.0018±0.0003 | 0.0023±0.0005 | 0.0028±0.0008 |
| protG | 0.0050±0.0013 | 0.0044±0.0009 | 0.0045±0.0015 | 0.0044±0.0011 | 0.0052±0.0019 |
| ntl9 | 0.0067±0.0073 | 0.0067±0.0075 | 0.0065±0.0059 | 0.0050±0.0063 | 0.0034±0.0006 |
| bba | 0.0093±0.0008 | 0.0119±0.0052 | 0.0125±0.0032 | 0.0149±0.0050 | 0.0131±0.0011 |
| cspa | 0.0171±0.0042 | 0.0159±0.0022 | 0.0188±0.0045 | 0.0177±0.0038 | 0.0210±0.0024 |
| T0765 | 0.0072±0.0025 | 0.0092±0.0023 | 0.0098±0.0026 | 0.0112±0.0022 | 0.0132±0.0036 |
| ww | 0.0246±0.0054 | 0.0256±0.0030 | 0.0277±0.0047 | 0.0304±0.0065 | 0.0326±0.0063 |
| T0855 | 0.0185±0.0024 | 0.0244±0.0060 | 0.0218±0.0015 | 0.0237±0.0026 | 0.0249±0.0028 |
| T0773 | 0.0125±0.0081 | 0.0162±0.0082 | 0.0167±0.0061 | 0.0183±0.0048 | 0.0202±0.0109 |
| T0816 | 0.0349±0.0038 | 0.0355±0.0051 | 0.0354±0.0052 | 0.0342±0.0024 | 0.0349±0.0021 |
| gpW | 0.0210±0.0112 | 0.0156±0.0055 | 0.0177±0.0025 | 0.0164±0.0035 | 0.0177±0.0058 |
| bbl | 0.0154±0.0058 | 0.0255±0.0159 | 0.0303±0.0172 | 0.0228±0.0060 | 0.0399±0.0363 |
| hyp | 0.0464±0.0175 | 0.0390±0.0062 | 0.0435±0.0070 | 0.0534±0.0080 | 0.0541±0.0077 |
| T0769 | 0.0072±0.0015 | 0.0105±0.0051 | 0.0081±0.0011 | 0.0102±0.0035 | 0.0106±0.0041 |
| T0771 | 0.0095±0.0014 | 0.0095±0.0012 | 0.0108±0.0024 | 0.0123±0.0029 | 0.0132±0.0021 |

Table 8: The future prediction results on NC

| Future frames | 10 | 11~20 | 21~30 | 31~40 | 41~50 |
|---|---|---|---|---|---|
| YiiP | 0.8752±0.0057 | 0.8505±0.0067 | 0.8226±0.0087 | 0.8269±0.0104 | 0.8249±0.0117 |
| NahA | 0.7863±0.0182 | 0.7854±0.0146 | 0.7564±0.0151 | 0.7513±0.0072 | 0.7400±0.0118 |
| adk_equi | 0.7750±0.0205 | 0.7723±0.0198 | 0.7329±0.0441 | 0.7514±0.0294 | 0.7529±0.0408 |
| protG | 0.8184±0.0514 | 0.8018±0.0580 | 0.8063±0.0376 | 0.7879±0.0653 | 0.7795±0.0705 |
| ntl9 | 0.7595±0.1691 | 0.8099±0.1178 | 0.8347±0.0429 | 0.7569±0.1273 | 0.7618±0.1284 |
| bba | 0.7259±0.0680 | 0.7323±0.0585 | 0.7703±0.0338 | 0.7344±0.0587 | 0.7122±0.0832 |
| cspa | 0.7477±0.0714 | 0.7243±0.0472 | 0.7040±0.0384 | 0.7242±0.0370 | 0.6859±0.0511 |
| T0765 | 0.6863±0.0664 | 0.6732±0.0648 | 0.6730±0.0410 | 0.6556±0.0360 | 0.6409±0.0258 |
| ww | 0.6313±0.1016 | 0.5828±0.0755 | 0.5697±0.0553 | 0.5362±0.0355 | 0.5386±0.0339 |
| T0855 | 0.6896±0.0400 | 0.6516±0.0394 | 0.6340±0.0306 | 0.6104±0.0240 | 0.5751±0.0377 |
| T0773 | 0.6588±0.0668 | 0.6393±0.0581 | 0.6551±0.0517 | 0.6422±0.0591 | 0.6219±0.0396 |
| T0816 | 0.7513±0.0869 | 0.7966±0.0635 | 0.7647±0.0790 | 0.7393±0.1030 | 0.7151±0.1231 |
| gpW | 0.6905±0.0715 | 0.6998±0.1042 | 0.7097±0.1046 | 0.7173±0.0746 | 0.6836±0.0724 |
| bbl | 0.5752±0.1445 | 0.6645±0.1352 | 0.6805±0.1300 | 0.5444±0.0363 | 0.5292±0.0266 |
| hyp | 0.6558±0.0692 | 0.6538±0.0722 | 0.6247±0.0652 | 0.5840±0.0506 | 0.6105±0.0493 |
| T0769 | 0.6662±0.0673 | 0.6734±0.0623 | 0.6193±0.0560 | 0.6527±0.0661 | 0.6723±0.0489 |
| T0771 | 0.6136±0.0458 | 0.5941±0.0373 | 0.5845±0.0322 | 0.5843±0.0304 | 0.5758±0.0266 |
| Future frames | 51~60 | 61~70 | 71~80 | 81~90 | 91~100 |
| YiiP | 0.8044±0.0084 | 0.7786±0.0117 | 0.7733±0.0047 | 0.7707±0.0100 | 0.7531±0.0058 |
| NahA | 0.7493±0.0103 | 0.7323±0.0052 | 0.7195±0.0084 | 0.7164±0.0094 | 0.7007±0.0095 |
| adk_equi | 0.7889±0.0126 | 0.8052±0.0162 | 0.7931±0.0085 | 0.7735±0.0222 | 0.7519±0.0262 |
| protG | 0.7754±0.0330 | 0.7838±0.0258 | 0.7910±0.0283 | 0.7810±0.0359 | 0.7608±0.0485 |
| ntl9 | 0.7378±0.1167 | 0.7330±0.1219 | 0.7148±0.0991 | 0.7658±0.0919 | 0.7753±0.0177 |
| bba | 0.7637±0.0127 | 0.7291±0.0683 | 0.7218±0.0528 | 0.6993±0.0607 | 0.7187±0.0136 |
| cspa | 0.6714±0.0448 | 0.6660±0.0248 | 0.6464±0.0425 | 0.6452±0.0436 | 0.5939±0.0337 |
| T0765 | 0.6476±0.0167 | 0.6218±0.0246 | 0.6156±0.0297 | 0.5940±0.0272 | 0.5774±0.0203 |
| ww | 0.5268±0.0294 | 0.5466±0.0283 | 0.5425±0.0231 | 0.5261±0.0206 | 0.5092±0.0121 |
| T0855 | 0.5920±0.0276 | 0.5566±0.0309 | 0.5737±0.0070 | 0.5505±0.0204 | 0.5432±0.0179 |
| T0773 | 0.6469±0.0345 | 0.6179±0.0404 | 0.6003±0.0373 | 0.5855±0.0301 | 0.5663±0.0235 |
| T0816 | 0.7629±0.0908 | 0.7722±0.0919 | 0.7296±0.1131 | 0.7657±0.0317 | 0.7070±0.0845 |
| gpW | 0.6158±0.1008 | 0.6914±0.0787 | 0.6581±0.0503 | 0.6627±0.0693 | 0.6653±0.0590 |
| bbl | 0.5579±0.0543 | 0.5243±0.0473 | 0.5101±0.0246 | 0.5204±0.0239 | 0.5127±0.0109 |
| hyp | 0.5703±0.0607 | 0.6011±0.0231 | 0.5737±0.0409 | 0.5576±0.0250 | 0.5596±0.0243 |
| T0769 | 0.6372±0.0511 | 0.6165±0.0432 | 0.6206±0.0423 | 0.6016±0.0355 | 0.6141±0.0253 |
| T0771 | 0.5688±0.0278 | 0.5505±0.0230 | 0.5495±0.0181 | 0.5367±0.0195 | 0.5339±0.0176 |

## E.3 GPCRmd

Table 9 shows the evaluations of the four proteins in GPCRmd on the three tasks of reconstruction, time interpolation and prediction of the future 50 frames. From the table, the results of different proteins are comparable, indicating that the model has a certain multi-protein generalization ability.

Table 9: Evaluation on GPCDmd

| | IoU↑ | Chamfer_dist↓ | NC↑ | IoU↑ | Chamfer_dist↓ | NC↑ |
|---|---|---|---|---|---|---|
| | | 10792_trj_81 | | | 10912_trj_95 | |
| Reconstruction | 0.8158±0.0224 | 0.0005±0.0001 | 0.8668±0.0242 | 0.8181±0.0242 | 0.0004±0.0002 | 0.8674±0.0242 |
| Interpolation | 0.7885±0.0173 | 0.0006±0.0001 | 0.8118±0.0159 | 0.7928±0.0180 | 0.0005±0.0001 | 0.8127±0.0176 |
| Future 50 frames | 0.7234±0.0317 | 0.0014±0.0005 | 0.7433±0.0319 | 0.7027±0.0502 | 0.0011±0.0003 | 0.7092±0.0544 |
| | | 15711_trj_791 | | | 16105_trj_848 | |
| Reconstruction | 0.7621±0.0508 | 0.0009±0.0004 | 0.8217±0.0462 | 0.7725±0.0516 | 0.0009±0.0004 | 0.8327±0.0476 |
| Interpolation | 0.7090±0.0556 | 0.0012±0.0005 | 0.7082±0.0520 | 0.7186±0.0623 | 0.0012±0.0006 | 0.7216±0.0604 |
| Future 50 frames | 0.5654±0.0762 | 0.0027±0.0011 | 0.5926±0.0485 | 0.6076±0.0462 | 0.0018±0.0007 | 0.6336±0.0389 |