# OpenReview forum: "DSR: Dynamical Surface Representation as Implicit Neural Networks for Protein"
_NeurIPS.cc/2023/Conference — NeurIPS 2023 poster_

### Official Review · Reviewer_Dvrt · 2023-07-04

**Soundness:** 3 good
**Presentation:** 3 good
**Contribution:** 3 good
**Rating:** 5
**Confidence:** 4

**Summary:**

The work proposes to model the surface of proteins using signed distance field (SDF) expressed by implicit neural representations. Previously, protein surfaces were studied to characterize protein interactions and ligand binding. In this work, surface representation learning is for analyzing MD simulation data. The network takes in 3D coordinates, as well as time, and returns SDF and is optimized in the typical fashion by minimizing the distance of 3D points on the surface, matching normals, plus Eikonal term regularization. The experiments on curated datasets show that network is able to provide dynamic reconstructions, while the performance degrades on structures that exhibit rapid movements. The networks is able to interpolate between moments and predict future dynamics.

**Strengths:**

1. The work is very-well written. In the introduction section, the relation with previous work and their issues are clarified and the task addressed in the work is well-motivated. Datasets used for evaluation are clearly described improving reproducibility.
2. Additional visualizations provide in the supplement and main text helps understanding results on modeling dynamics with implicit neural representations. For example, both quantitative and qualitative results shows that with increase of RMSD the reconstructions do not match well with the ground-truth.
3. The proposed method is evaluated in multiple contexts. For example, its ability to interpolate or extrapolate in time is assessed. Also, a case study shows that one can learn a latent space encode dynamics of similar proteins.


**Weaknesses:**

1. According to the provided visualizations (e.g. Figure 3), the proposed method could add some non-existent tiny areas close to the structure but disconnected. I believe these artifacts are due to implicitly representing surfaces.
2. In L211, it is mentioned that the reconstructions are clearer than GT. This is a weird statement as one expect that GT be the most faithful representation of the surface. It is not clear if small rapid movements are due to noise or part of the true dynamics; If the latter, then the inductive bias in the network to output a smooth function is not really a merit of the work. On the other hand, if the GT is noisy, then it is not clear how to quantitatively evaluate the reconstructions; thus such results become less valid and hard to rely on.
3. It is claimed that, unlike previous work, the proposed method is able to resolve dynamics of large proteins. However, there is no quantitative comparison with previous work to validate this.
4. It is not clearly stated why implicit representation for surfaces is used, rather than other representations (such as algebraic ones). There is a lack of experimental validation on this, too.

**Questions:**

1. I suggest that you provide more context on why those rapid movements in the structure is noise and not part of real dynamics. I think that it is not fully noise but a blend of noise and true dynamics which is completely eliminated by smoothness inductive bias in the network.
2. In the interpolation experiments, how do you extract one frame out of several frames? How much is the RMSD between two states that are used as endpoints of interpolation?

**Limitations:**

Please address the weaknesses and questions above.

---

> ### Author Rebuttal · Authors · 2023-08-09
>
> **Q1**: According to the provided visualizations (e.g. Figure 3), the proposed method could add some non-existent tiny areas close to the structure but disconnected. I believe these artifacts are due to implicitly representing surfaces.
>
> **A1**: We acknowledge the complexity of protein surfaces, which pose significant challenges in characterizing them using a continuous function model. Our study focuses on a continuous function model in a 3D+time space, which increases the learning difficulty. In Figure 3, the protein exhibits substantial dynamic variations, including transitions between different folding patterns. This adds to the complexity of learning across both spatial and temporal dimensions. In comparison, proteins with smaller motion amplitudes or more regular oscillations tend to exhibit fewer of these tiny areas.
>
> **Q2**: The statement that the reconstructions are clearer than GT may not be appropriate.
>
> **A2**: Here, we would like to clarify that the surface representation of proteins that we have access to is in the form of discrete point clouds, which inherently have limited resolution. We aim to use a continuous model to capture a more smooth representation. Perhaps using the term "GT" might not be the most appropriate in this context, and "reference" would be a more suitable term.
>
> **Q3**: It is claimed that, unlike previous work, the proposed method is able to resolve dynamics of large proteins. However, there is no quantitative comparison with previous work to validate this.
>
> **A3**: In response to this concern, we would like to provide a comprehensive explanation of the computational complexity of traditional methods for molecular dynamics simulation. For instance, solving the Schrödinger equation directly for a system of 100 atoms would require solving a matrix with approximately $1.26 \times 10^6$ elements. As the number of atoms increases, the computational intractability of this method becomes evident, even with the aid of supercomputers. Additionally, Density Functional Theory (DFT), a widely used approximation for solving the Schrödinger equation, has a time complexity of $O(n^3)$, where n represents the number of atoms. Consequently, a DFT calculation for a system of 1000 atoms would cost around 1 billion calculations. In contrast, our method, which converts the protein representation to a continuous function of the protein surface, significantly reduces the computational complexity. Unlike traditional methods that heavily rely on atomic representations of proteins, our approach for generating the dynamic representation of proteins is independent of the number of atoms. With a well-trained model, the generation of each time step's surface structure, on average, only requires approximately 30 seconds.
>
> **Q4**: It is not clearly stated why implicit representation for surfaces is used, rather than other representations (such as algebraic ones). There is a lack of experimental validation on this, too.
>
> **A4**: We appreciate your concerns. In our response, we would like to emphasize the importance of the protein surface and why we chose the implicit neural representation method. Traditionally, biochemists have used atomic representations to describe proteins. However, the most accurate methods for molecular dynamics simulations rely on quantum mechanics. From a quantum mechanics perspective, both atoms and electrons have inherent uncertainties. Therefore, using atomic representations to calculate atomic forces may not be accurate, and representing proteins through electron density maps might be a better choice.
>
> Hence, in order to model the dynamics of proteins from this perspective, we initially chose to represent the protein surface. Implicit neural representation has proven to be an excellent method for modeling 3D objects. However, we acknowledge that we did not provide experimental validation for this specific choice.
>
> Regarding the comparison with other methods, as we are the first to use this approach to model such problems, there is no previous work to compare against. However, based on the suggestions from other reviewers, we have conducted additional comparison experiments with linear interpolation methods to demonstrate that our approach can indeed capture the dynamic changes of protein surfaces. The complete results table (Table 1. and Table 2. ) and comparative charts (Figure 1. ) can be found in the attachment PDF in the global rebuttal, and we have included the results and a comparison line chart in the revised manuscript.
>
> **Q5**: Why are those rapid movements in the structure noise and not part of real dynamics?
>
> **A5**: Thank you for your suggestion. Indeed, the rapid movements in the protein structure contain both noise and true dynamics. However, what I intended to convey was the smoothness of the trajectory. In other words, it refers to the ease or difficulty of modeling that particular case.
>
> The continuous inductive bias of the model does help to eliminate noise to some extent. However, our evaluation of the model is based on the data we have access to. The application of smoothing techniques may lead to a decrease in evaluation metrics. At present, it is challenging to determine which representation is more appropriate. In our future work, we plan to quantitatively evaluate the performance of the representations through downstream tasks.
>
> **Q6**: In the interpolation experiments, how do you extract one frame out of several frames? How much is the RMSD between two states that are used as endpoints of interpolation?
>
> **A6**: In our modeling approach, we extracted frames at fixed time intervals, specifically selecting every 10th frame to ensure equal time intervals between the frames. We did not extract frames based on the RMSD between states. This decision was made because our aim was to model a continuous function of 3D+time, and thus, we provided the model with a time-related input to facilitate the learning of temporal dynamics.

---

> > ### Comment · Reviewer_Dvrt · 2023-08-21
> >
> > Thank you for the detailed response. I liked the discussion around the practical/computational benefit for large proteins.
> > Yet, some of the concerns are not addressed so I rather keep my score. As authors agreed that rapid motions of the protein structure could contain both noise and true dynamics, namely the reference data is itself noisy, it is not clear how to reliably validate the reconstructions. This is limitations which is not discussed in the work. Also, further evaluations on downstream tasks are missing at current stage.

---

### Official Review · Reviewer_MFaL · 2023-07-06

**Soundness:** 3 good
**Presentation:** 3 good
**Contribution:** 3 good
**Rating:** 5
**Confidence:** 3

**Summary:**

This paper proposes to learn an implicit neural network to model protein surface and dynamics. A protein surface is represented as a signed distance function (SDF), which can be modeled by implicit neural networks based on MLP. The training objective follows standard practice: the difference between true and predicted SDF values at each point (both interior and exterior) and an Eikonal term at the surface points to ensure the learned function is a valid SDF. DSR is evaluated on multiple protein dynamic simulation data such as GPCRmd. The model is able to reconstruct the surface with different levels of accuracy, depending on the types of proteins. DSR is also able to predict the future frames given partial protein trajectory.

**Strengths:**

* This paper presents an interesting direction to protein surface. Modeling protein dynamics at the surface level is a novel contribution to the field.
* Using implicit neural networks to model protein surface is a novel application of existing surface modeling methods.
* This paper presents a comprehensive analysis over a collection of 20 protein dynamic datasets, which can be a valuable resource to the protein dynamic modeling field.

**Weaknesses:**

There are two limitation of the method.
1. Currently, DSR cannot generalize to a new protein. In the current set of results, DSR is trained and tested on the same protein, i.e., it is trained on a partial trajectory of one protein and used to predict the future of that protein. In practice, we usually want to simulate a protein that we haven't simulated before.
2. DSR only models the surface of the protein. Although the shape of a protein contains useful information, it is not all. We also need to know the chemical composition over the protein surface, i.e., which part of the surface is more hydrophobic/charged/etc. Predicting the surface itself is not sufficient for downstream analysis. For example, MaSIF (Nature Method 2020) used protein surface to predict protein-protein interaction. They find that their model performs poorly when using only the geometric shape as input. When they add chemical features of the protein surface patches (hydrophobicity, charge, etc.), the performance improved substantially. Therefore, it is necessary to predict not only the geometric shape but also the chemical composition of the protein surface.

**Questions:**

In section 4.3.3, when you model multiple proteins, do you provide protein type as an input to DSR model? If not, how do you train the model so that DSR knows there are different protein types?

**Limitations:**

The paper has included a short discussion on limitations and societal impact.

---

> ### Author Rebuttal · Authors · 2023-08-09
>
> Thanks for your review comments.
>
> **Q1**: Currently, DSR cannot generalize to a new protein.
>
> **A1**: We appreciate your valuable feedback regarding the generalizability of our DSR method to new proteins. We acknowledge that the current set of results demonstrates a limitation in the ability of DSR to generalize to new proteins, which is a concern for practical applications.
>
> However, it is crucial to emphasize the significance of our work in establishing the foundation for continuous modeling of protein surfaces based on implicit neural representations. Without achieving continuous modeling of protein surfaces, it would be challenging to make progress in this field. By demonstrating the effectiveness of DSR in predicting the future dynamics of a protein based on a partial trajectory of the same protein, we provide a starting point for further investigations.
>
> It is important to note that our research is still in the exploratory stage, and we are actively working on improving the practicality of our method. In our upcoming work, we plan to address the issue of generalizability by introducing and learning a vector field (force field) that can simulate the dynamics of new proteins given an initial structure. This extension will enable our model to simulate the dynamics of new proteins based on the learned force field.
>
> We appreciate the reviewer's insightful comments, and we are dedicated to addressing these challenges in our ongoing research.
>
> **Q2**: DSR only models the surface of the protein. Although the shape of a protein contains useful information, it is not all. We also need to know the chemical composition over the protein surface, i.e., which part of the surface is more hydrophobic/charged/etc. Predicting the surface itself is not sufficient for downstream analysis.
>
> **A2**: Indeed, it is an active area of research in the community to explore the physical and chemical properties of protein surfaces for downstream tasks such as protein-protein interactions (PPI) and docking. However, our work focuses on addressing some of the underlying and underappreciated aspects of protein representation.
>
> Our motivation stems from the fact that traditional molecular simulations are based on quantum mechanics, where atoms and electrons have inherent uncertainties. In this regard, we believe that representing proteins using electron density maps, as opposed to atomic representations, may be more suitable. Therefore, we adopted a surface-based representation of proteins to capture their shape. While geometric information is the initial step, we fully recognize the importance of incorporating chemical and biological information to enhance our model and expand its applicability.
>
> In our ongoing research, we are actively working on incorporating chemical and biological features to enrich our model. We aim to explore and integrate additional information such as hydrophobicity, charge, and other relevant properties of the protein surface. By doing so, we intend to strengthen our model's capability and enable it to be applied to a wider range of downstream tasks.
>
> We appreciate the reviewer's insightful comment and will continue to improve our approach by incorporating chemical composition and exploring various downstream applications.
>
> **Q3**: In section 4.3.3, when you model multiple proteins, do you provide protein type as an input to DSR model? If not, how do you train the model so that DSR knows there are different protein types?
>
> **A3**: In our model, we introduce a latent code, which is a vector with learnable parameters. By specifying an index for the latent code, we inform the model about the specific protein type. This allows the DSR model to differentiate between different protein types during training and prediction.
>
> Furthermore, we acknowledge the potential for further exploration of the latent space. We aim to collect additional data in the future to investigate the generalization capabilities of our model across a broader range of protein types. By expanding the dataset, we hope to enhance the model's ability to generalize to different protein types. Thank you for raising this important question, and we appreciate the opportunity to clarify the mechanism by which we incorporate protein type information into our DSR model.

---

> > ### Comment · Reviewer_MFaL · 2023-08-15
> > **Thank you for your response**
> >
> > Dear authors,
> >
> > Thank you for your detailed response. I understand that the current method is still in exploratory stage and the direction is interesting and promising. However, it is still unclear if the model can be used to simulate dynamics of a new protein. Is it possible to conduct the generalization experiment using the multi-GPCR dataset? E.g., train the model on a subset of GPCR dynamics and test it on others?

---

> > > ### Author Response · Authors · 2023-08-16
> > > **Response to Reviewer MFaL**
> > >
> > > Thank you for your response and comments.
> > >
> > > The goal of our work is to validate the use of a surface model to represent the dynamics of proteins. This surface model is built upon a continuous representation of time and space. The issue of generalization you mentioned, which refers to the ability to simulate on new proteins, is a concern for us, but it is not the main focus of this work.
> > >
> > > The current approach, based on implicit neural representations, aims to find a more appropriate and meaningful understanding of protein manifold representation. For structural biologists, in most cases, they do not need to study multiple proteins simultaneously but rather want to discover more possibilities within a single protein. For example, in the case of GPCRs, a large class of proteins that are highly similar in sequence and structure, there may be various dynamic changes that are challenging to simulate using traditional molecular dynamics approaches. In our model design, we introduce a latent code that enables the model to capture the motion patterns of a specific class of proteins. In the future, we plan to explore the latent space formed by the latent codes to enable the model to generate reasonable and diverse protein dynamic variations, which means the model may have generation ability.
> > >
> > > Furthermore, as a temporal model for protein surface representation, our model exhibits strong expressive power and good temporal and spatial interpolation capabilities. The current model effectively captures the temporal information of protein dynamics.

---

> > > ### Author Response · Authors · 2023-08-20
> > > **Another Response to Reviewer MFaL**
> > >
> > > We would like to kindly inquire if you were satisfied with our previous response. We greatly appreciate your question regarding the utilization of our model on new proteins. Regarding your query about the possibility of conducting generalization experiments on the GPCR dataset, we believe that our model has strong potential to explore new conformational spaces within a specific class of proteins, such as GPCRs, which share similar structural configurations. It is feasible to explore the diversity of dynamics by leveraging latent space encoded through latent codes. However, achieving generalization across different proteins currently poses challenges. This challenge is also faced by implicit neural representation models in characterizing 3D objects. For example, prominent neural representation models like NeRF [1], NeuS [2], and IGR[3] struggle to achieve generalization, despite their pioneering contributions that have shaped the advancement of this field. In contrast, our work's primary contribution lies in proposing a more suitable and efficient representation of protein surface dynamics.
> > >
> > > [1] Mildenhall B, Srinivasan P P, Tancik M, et al. Nerf: Representing scenes as neural radiance fields for view synthesis[J]. Communications of the ACM, 2021, 65(1): 99-106.
> > >
> > > [2] Wang P, Liu L, Liu Y, et al. Neus: Learning neural implicit surfaces by volume rendering for multi-view reconstruction[J]. arXiv preprint arXiv:2106.10689, 2021.
> > >
> > > [3] Gropp A, Yariv L, Haim N, et al. Implicit geometric regularization for learning shapes[J]. arXiv preprint arXiv:2002.10099, 2020.

---

> > > > ### Comment · Reviewer_MFaL · 2023-08-21
> > > > **Thank you for your response**
> > > >
> > > > I think the current method is still in exploratory stages. Despite some limitations, this direction is promising and interesting so I will raise my score in agreement with other reviewers.

---

### Official Review · Reviewer_EXzZ · 2023-07-11

**Soundness:** 3 good
**Presentation:** 3 good
**Contribution:** 3 good
**Rating:** 5
**Confidence:** 3

**Summary:**

The author puts forwards the DSR model, which is an implicit neural network, uses this model to accurately capture the protein dynamic trajectories and interpolate and extrapolate in 3D and time. The DSR is a scalable and efficient method to modeling protein dynamics, overcomes the limitations of first-principles-based and deep learning methods. As the most important contribution, it is successful for the large protein dynamic study.

**Strengths:**

1. The method learns SDF well and achieves dynamic modeling for large proteins for the first time. This is an excellent strength.
2. The surface representation approach is useful in protein dynamic field. It simplifies calculations and allows identifying movement trends and amplitudes of protein domains.

**Weaknesses:**

1. In the reconstruction task, there are little comparison between the DSR and other methods.
2.

**Questions:**

1. In the background, there are some traditional method which is costly and low speed. Is there some reason or explanation that this method is fast and cheap, other than the application of neural network?
2. In the reconstruction task, there are three evaluation metrics, in what range do the values of these metrics indicate that the model is performing well?


**Limitations:**

1. The DSR model is sensitive to noise in protein dynamic trajectories, which is a challenge while faced with data with little smoothness or great noise.
2. As the analysis for the GPCRmd, the DSR model has the potential to learn the dynamics of a class of similar proteins, such as a protein family.

---

> ### Author Rebuttal · Authors · 2023-08-09
>
> Thanks for your review comments.
>
> **Q1**: In the reconstruction task, there are little comparison between the DSR and other methods.
>
> **A1**: Thank you for your feedback regarding the comparison between our DSR method and other methods in the reconstruction task. As we are the first to propose and model this issue, there are no previous works to compare our method with. However, at the suggestion of another reviewer, we have added a comparison with a linear interpolation to our manuscript. A partial summary of the results is shown in the table below. The complete results table (Table 1. and Table 2. ) and comparative charts (Figure 1. ) can be found in the attachment PDF in the global rebuttal, and we have included the results and a comparison line chart in the revised manuscript.
>
> **Table: Comparison of evaluation on the surface interpolation between trained frames by our model and linear interpolation**
>
> |||Ours||||Linear||
> |:-:|:-:|:-:|:-:|-|:-:|:-:|:-:|
> ||IoU|Chamfer_dist|NC||IoU|Chamfer_dist|NC|
> |DIMS|0.9225±0.0035|0.0003±0.0001|0.9527±0.0034||0.9084±0.0071|0.0003±0.0001|0.9266±0.0079|
> |I-FABP|0.8833±0.0078|0.0006±0.0001|0.9254±0.0066||0.8770±0.0076|0.0006±0.0001|0.9053±0.0082|
> |adk_equi|0.7459±0.0436|0.0021±0.0012|0.7816±0.0327||0.7181±0.0440|0.0020±0.0010|0.7320±0.0352|
> |bba|0.6204±0.1296|0.0052±0.0064|0.8074±0.0972||0.5249±0.1506|0.0076±0.0079|0.7193±0.1056|
> |cspa|0.5957±0.0714|0.0080±0.0039|0.7812±0.0574||0.5189±0.0728|0.0070±0.0032|0.6994±0.0556|
> |T0816|0.5245±0.1289|0.0087±0.0088|0.7221±0.0951||0.3994±0.1344|0.0097±0.0092|0.6259±0.0845|
> |gpW|0.4769±0.1262|0.0097±0.0086|0.7008±0.0967||0.3846±0.1411|0.0109±0.0092|0.6228±0.0912|
> |bbl|0.4380±0.1372|0.0108±0.0088|0.6605±0.0895||0.3304±0.1558|0.0130±0.0122|0.5964±0.0881|
>
> The comparison results demonstrate that our method outperforms the linear interpolation method, indicating that our model is capable of learning the inherent dynamic changes of proteins. We have updated the manuscript to reflect this comparison.
>
> **Q2**: In the background, there are some traditional method which is costly and low speed. Is there some reason or explanation that this method is fast and cheap, other than the application of neural network?
>
> **A2**: Thank you for your feedback The traditional methods for molecular dynamics simulation, such as solving the Schrödinger equation, the Born-Oppenheimer approximation method, and Density Functional Theory (DFT), are computationally expensive and time-consuming. To provide a more detailed explanation, we can give specific numbers to illustrate the computational complexity of these methods. For instance, solving the Schrödinger equation directly for a system of 100 atoms would require solving a matrix with approximately $1.26 \times 10^6$ elements, which is computationally intractable even with supercomputers. The Born-Oppenheimer approximation, which separates electronic and nuclear motion, is also computationally expensive. For example, a single energy minimization calculation using this approximation for a protein containing 10,000 atoms can take several hours on a high-performance computing cluster. DFT, which is a widely used approximation for solving the Schrödinger equation, has a time complexity of $O(n^3)$, where $n$ is the number of atoms in the system. For example, a DFT calculation for a system of 1000 atoms would require approximately 1 billion calculations.
>
> Since traditional biochemists tend to focus on atomic representations of proteins, the computational complexity is closely related to the number of atoms in the system. However, we have converted the representation to a continuous function of the protein surface, which can significantly reduce the computational complexity.
>
> **Q3**: In the reconstruction task, there are three evaluation metrics, in what range do the values of these metrics indicate that the model is performing well?
>
> **A3**: In general, for the IoU metric, a value of 0.5 or above is commonly considered as a good performance indicator. This signifies a significant overlap between the predicted and ground truth structures. However, it's worth noting that specific assessment criteria may vary depending on the application domain and the particular task at hand.
>
> Since there is no established performance score for the problem we are studying, we have provided a comparison table (Table 2) of evaluation metrics for a 3D reconstruction task as a reference for reviewers to understand[1]. Additionally, we have included visualizations in the figures and videos in the provided link https://anonymouswk.github.io/DSR to showcase the excellent modeling of protein surface dynamics achieved by our method.
>
> **Table2: Single view reconstruction results on Pix3D Chairs.**
>
> |Metric|Pix3D|AtlasNet|Mesh R-CNN|Pixel2Mesh|DISN|MeshSDF (raw)|MeshSDF|
> |:-:|:-:|:-:|:-:|:-:|:-:|:-:|:-:|
> |IoU|0.282|-|0.240|0.254|0.333|0.337|0.407|
> |EMD|0.118|0.128|0.125|0.115|0.117|0.119|0.098|
> |CD-$\sqrt{l_2}$|0.119|0.125|0.110|0.104|0.104|0.102|0.089|
>
> It is important to emphasize that determining whether a representation is good or not ultimately depends on its impact on downstream tasks. Further extending such representations to downstream work is an ongoing effort for us. We are actively working on integrating our approach into downstream tasks to assess its effectiveness and applicability.
>
> [1] Remelli E, Lukoianov A, Richter S, et al. Meshsdf: Differentiable iso-surface extraction[J]. Advances in Neural Information Processing Systems, 2020, 33: 22468-22478.

---

### Official Review · Reviewer_ZJGr · 2023-07-26

**Soundness:** 3 good
**Presentation:** 2 fair
**Contribution:** 2 fair
**Rating:** 5
**Confidence:** 4

**Summary:**

The paper proposes a framework for predicting protein dynamics through learning the time-dependent SDF. The authors showcase that the learned SDF can reconstruct/interpolate protein surfaces and predict the dynamics, thus bypassing the computationally expensive molecular dynamics simulation.

**Strengths:**

* The paper validates the model by conducting extensive numerical experiments on various protein systems.
* Learning efficient surrogate models for protein dynamics prediction is important due to the computational cost associated with simulating protein dynamics using MD. The proposed method can potentially benefit both computational biology and drug discovery efforts.

**Weaknesses:**

The presentation of the current manuscript can make it difficult to grasp its technical contribution. From the perspective of developing new deep learning models, the paper has combined many existing techniques. For example, parametrizing SDF with neural networks have extensively studied in many neural field works like SIREN/DeepSDF, and the Eikonal term has also been explored in prior work IGR (Gropp, Amos, et al. ). The experiments also do not show how it addresses or improves upon issues in existing deep learning models. For example, what is the advantage of the proposed method over a point-cloud GNN that learns to predict atom-wise displacement?

On the other hand, if the authors' primary objective is to introduce a new way to predict the protein dynamics for the simulation community, it is also not very clear to what extent the model can be used to replace classical MD. To use the model for prediction, whenever there is a new protein system, one needs to first train the model on its existing trajectory (needs to run MD to get it) and then extrapolate from it. The model cannot do any prediction from scratch on new configuration of molecular system without re-training.

In addition, the model is neither energy conserved nor equivariant, this might not be a serious issue in fitting and reconstructing a single protein trajectory but can cause instability for the long-term prediction.

**Questions:**

* The units for RMSD and CD in the result section are missing (I assume it is Angstrom).
*  The latent code "z" in the equation 2 is not defined and discussed in the method section.

**Limitations:**

The authors have discussed the limitation of their work in the conclusion section.

---

> ### Author Rebuttal · Authors · 2023-08-09
>
> Thanks for your review comments.
>
> **Q1**: What is the advantage of the proposed method over a point-cloud GNN that learns to predict atom-wise displacement?
>
> **A1**: Thank you for your question regarding the advantages of our proposed method over a point-cloud GNN that learns to predict atom-wise displacement. While our work is inspired by previous works such as DeepSDF and IGR, our approach is based on the claim that a protein's surface is crucial for understanding its properties.
>
> Traditional biochemists tend to focus on atomic representations of proteins. However, from a quantum mechanics perspective, atoms and electrons have inherent uncertainties. Therefore, representing proteins using electron density maps to study their properties may be more appropriate, which is why we chose to focus on protein surface representations.
>
> Based on our work, we are also conducting further research, including issues related to the generalization of larger protein spaces, modeling protein surface force fields based on vector fields, and assisting protein simulation from scratch.
>
> Regarding the advantages over point-cloud GNN, we claim that using a continuous surface representation can better cover uncertainties compared to a discrete atomic representation.
>
> **Q2**: It is also not very clear to what extent the model can be used to replace classical MD.
>
> **A2**: Thank you for your feedback regarding the extent to which our model can replace classical molecular dynamics (MD) for predicting protein dynamics. We understand the need for step-by-step progress towards achieving the goal of replacing MD. To address this limitation and pave the way for future advancements, we have introduced a latent code in our model, which allows for greater generalization across various protein types. This latent code serves as a starting point for training the model on existing trajectory data obtained from MD simulations. This paper is just the beginning of our work to validate the feasibility of using a protein surface's implicit neural representation. We are currently developing a follow-up study that incorporates and learns a vector field to guide proteins in simulating from scratch. We would like to emphasize that our paper represents an initial exploration into the feasibility of using a protein surface's implicit neural representation. We recognize the need for further research and development to bridge the gap between our current approach and a fully independent prediction model.
>
> **Q3**: The units for RMSD and CD in the result section are missing (I assume it is Angstrom).
>
> **A3**: Thank you for your feedback regarding the missing units for RMSD and CD in the results section. We confirm that the unit for RMSD is indeed Angstrom, which has been added to the manuscript. For Chamfer Distance (CD), this is a standardized unitless quantity with values ranging from 0 to 1. We have added an explanation of this to the manuscript. Thank you for bringing these issues to our attention, and we appreciate your helpful feedback.
>
> **Q4**: The latent code $z$ in the equation 2 is not defined and discussed in the method section.
>
> **A4**: Thank you for your feedback regarding the latent code $z$ in Equation 2. Here, we explain the latent code and we have added the definition and discussion in the manuscript. As previously mentioned, the latent code is for the model's generalization, and its parameters are learned during training. In Section 4.3.3 of the manuscript, we demonstrate the use of different latent codes for each protein in our multiple protein dynamics.
>
> Actually, we claim that with sufficient data, we can explore the latent space to achieve greater generalization across various protein types. We apologize for any confusion caused and will revise the manuscript accordingly to provide a more detailed explanation.

---

> > ### Comment · Reviewer_ZJGr · 2023-08-13
> > **Reply to author's response**
> >
> > I would like to thank the authors for the response and I appreciate the efforts they have made during the rebuttal period. However, my main concerns still remain and I would like to keep my score for the current revision.
> >
> > The authors have claimed in the introduction that "*We have achieved dynamic modeling of large proteins by using surface representations,
> >  which overcomes the limitations of previous atomic models or coarse-grained models that were restricted to small molecules or proteins*", therefore it is important to provide quantitative analysis on the generalization/extrapolation capability of the latent code (even if the performance is not good). Otherwise, at this stage, the model cannot be considered as any surrogate to MD/coarse-grained MD as it cannot simulate any new system unless there is an existing trajectory. And the authors state that atomic models cannot scale to large systems but there are actually several examples that have applied learned atomic forcefield to systems with millions of atoms [1, 2].
> >
> > [1] Learning local equivariant representations for large-scale atomistic dynamics
> >
> > [2] Pushing the limit of molecular dynamics with ab-initio accuracy to 100 million atoms with machine learning

---

> > > ### Author Response · Authors · 2023-08-15
> > > **Response to Reviewer ZJGr**
> > >
> > > Thank you for your response.
> > >
> > > Firstly, in each of our experimental objects, we conducted experiments on the extrapolation ability and performed quantitative analysis based on evaluation metrics. Regarding the generalization capability of the latent code, we discussed it in the case study of GPCRmd. Since there is currently a lack of abundant and comprehensive molecular dynamics simulation data, we chose an important GPCR protein and studied the generalization based on the latent code across different GPCR families. The experimental results showed that our model can capture the main functional motion in each family.
> > >
> > > Furthermore, besides the traditional atomic representation, we wanted to explore the possibility of using surface representation to depict protein dynamics. Proteins, as biological macromolecules, differ from other small molecules in that people are more interested in their functions, properties, and whether they bind based on surface complementarity, among other factors. Surface representation may provide a more intuitive way to visualize these aspects. This is a system we are trying to establish, and it is not easy. Currently, our first step is to model protein dynamics using surface representation (not simulation), and the experimental results indicate its feasibility, enabling us to further explore other capabilities and generalizations. As reviewer MFaL mentioned, *"Modeling protein dynamics at the surface level is a novel contribution to the field."* Additionally, storing long molecular dynamics trajectories for large protein systems is also a challenge. The storage space is directly proportional to the length of time and the number of atoms. For example, a 90 ns trajectory of a protein system with ten thousand atoms occupies 3.5 GB of space, while our model only requires 22 MB.
> > >
> > > Finally, regarding the two papers you mentioned about learning atomic force fields for million-atom systems, I believe there might be a misunderstanding in this regard. The paper [1] actually models the dynamics of small molecules, and the experimental datasets used are QM9, MD-17, and the $Li_3PO_4$ system, which are relatively easy to model with around 10 atoms. The authors mention expanding the system to a large number of atoms (421,824 atoms), but this was specifically done for the $Li_3PO_4$ system, which contains a large number of repetitive units, and the forces acting on each atom from the surrounding atoms are similar and can be computed in parallel. The paper [2] is similar, studying water and metallic copper systems. These examples are clearly different from large-scale proteins, where performing such parallel computations for systems with a large number of atoms in proteins is not feasible. Based on our current knowledge, there is no deep learning method that can perform dynamic simulations of large-scale proteins, and even for small protein systems, it remains challenging.

---

> > > > ### Comment · Reviewer_ZJGr · 2023-08-15
> > > > **Reply to author's response**
> > > >
> > > > Thanks author for the clarification which alleviate some of my concerns, and I agree with author that modelling protein surface with neural field-like method indeed offers non-trivial advancement over other atomic GNN models in terms of scalability and efficiency. I've adjusted my score by one point.
> > > >
> > > >  However it is still not clear to me how the model can be used on new system. I assume that the GPCR experiment shown in the paper is done by training 1 network on 4 systems with 4 assigned latent codes and this network can learn to model all 4 of them. But if given another new configuration (not in the training systems), how should one get the latent code for this new system to use the trained network? I think randomly initialize another new latent code would definitely not work well.

---

> > > > > ### Author Response · Authors · 2023-08-16
> > > > > **Response to Reviewer ZJGr**
> > > > >
> > > > > We appreciate your valuable and insightful comments.
> > > > >
> > > > > Our current approach using implicit neural representations aims to find a more suitable and meaningful understanding of protein manifold representation. For structural biologists, in most cases, they do not need to study multiple proteins simultaneously but rather want to discover more possibilities within a single protein. For example, GPCRs, a large class of proteins, are highly similar in sequence and structure, but they may exhibit various dynamic changes that are challenging to simulate using traditional molecular dynamics approaches. In our model, we introduce latent codes to capture the motion patterns of a specific class of proteins while allowing for individual variations in the overall motion pattern. Currently, we sample other latent codes from a standard normal distribution to explore diverse protein dynamic variations. In the future, we plan to explore the latent space formed by the latent codes, enabling the model to have the generation capability for generating reasonable and diverse protein dynamic variations. And if you could provide some feasible experimental suggestions, we would greatly appreciate it.
> > > > >
> > > > > At the same time, the goal of our work is to validate the surface  representation of protein dynamics. This surface model is built upon a continuous representation of time and space. While generalization, simulating on new proteins, is a concern for us, but it is not the main focus of this work. As a temporal model for protein surface representation, our model exhibits strong expressive power and good temporal and spatial interpolation capabilities. The current model effectively captures the temporal information of protein dynamics.

---

### Official Review · Reviewer_2SKi · 2023-07-26

**Soundness:** 3 good
**Presentation:** 2 fair
**Contribution:** 4 excellent
**Rating:** 7
**Confidence:** 3

**Summary:**

The paper proposes to learn an implicit neural representation to model protein surface across continuous spatial and temporal space.

**Strengths:**

This is a great application of implicit neural networks for modeling protein surfaces, and the results seem promising.

**Weaknesses:**

The evaluation of the method can be improved:
- Even though there is no prior work as a baseline, simple methods should still be used to confirm the learned model is not doing something trivial. For instance, what happens if the author just linearly interpolate the 3D coordinates for the experiment in 4.3?
- The reconstructions metrics do not really reflect the usability of the model predicted trajectory. Can we see the correlation for docking on ground truth and generated surface? What about the distance between ground truth and the generated surface in terms of their MaSIF distance?

**Questions:**

I think the paper can be greatly improved if the author can address the weaknesses section and I am happy to consider raising my score then.

In addition, the author should consider spending more text to describe the evaluation setup, specifically around how train/test data is split and what has been done to prevent data leakage.

**Limitations:**

The author has sufficiently and fairly discussed the limitation of their methods.

---

> ### Author Rebuttal · Authors · 2023-08-09
>
> Thanks for your review comments.
>
> **Q1**: Comparing the learned model with simple methods, such as linearly interpolate?
>
> **A1**: Following the reviewer's suggestion, we conduct additional experiments comparing our methods to linear interpolation that generates intermediate structures between two endpoints.
>
> A partial summary of the results is shown in the table below. The complete results table (Table 1. and Table 2. ) and comparative charts (Figure 1. ) can be found in the attachment PDF in the global rebuttal, and we have included the results and a comparison line chart in the revised manuscript. As shown in the table below, our method outperforms linear interpolation in terms of surface quality. However, we would like to point out that this comparison is not entirely fair to our method, as linear interpolation skips the difficulty of generating reasonable surface details. In contrast, our model is capable of learning the underlying structure of the data and generating high-quality surfaces that capture the subtle details of the input data. We believe that these additional experiments and comparisons demonstrate the strengths of our method and its ability to generate high-quality surfaces that capture the complexity and nuances of the input data.
>
> **Table: Comparison of evaluation on the surface interpolation between trained frames by our model and linear interpolation**
>
> |  |  | Ours |  |  |  | Linear |  |
> |:---:|:---:|:---:|:---:|---|:---:|:---:|:---:|
> |  | IoU | Chamfer_dist | NC |  | IoU | Chamfer_dist | NC |
> | DIMS | 0.9225±0.0035 | 0.0003±0.0001 | 0.9527±0.0034 |  | 0.9084±0.0071 | 0.0003±0.0001 | 0.9266±0.0079 |
> | I-FABP | 0.8833±0.0078 | 0.0006±0.0001 | 0.9254±0.0066 |  | 0.8770±0.0076 | 0.0006±0.0001 | 0.9053±0.0082 |
> | adk_equi | 0.7459±0.0436 | 0.0021±0.0012 | 0.7816±0.0327 |  | 0.7181±0.0440 | 0.0020±0.0010 | 0.7320±0.0352 |
> | bba | 0.6204±0.1296 | 0.0052±0.0064 | 0.8074±0.0972 |  | 0.5249±0.1506 | 0.0076±0.0079 | 0.7193±0.1056 |
> | cspa | 0.5957±0.0714 | 0.0080±0.0039 | 0.7812±0.0574 |  | 0.5189±0.0728 | 0.0070±0.0032 | 0.6994±0.0556 |
> | T0816 | 0.5245±0.1289 | 0.0087±0.0088 | 0.7221±0.0951 |  | 0.3994±0.1344 | 0.0097±0.0092 | 0.6259±0.0845 |
> | gpW | 0.4769±0.1262 | 0.0097±0.0086 | 0.7008±0.0967 |  | 0.3846±0.1411 | 0.0109±0.0092 | 0.6228±0.0912 |
> | bbl | 0.4380±0.1372 | 0.0108±0.0088 | 0.6605±0.0895 |  | 0.3304±0.1558 | 0.0130±0.0122 | 0.5964±0.0881 |
>
> We believe that our model is effective in handling protein dynamic changes and has practical applications. We will continue to improve our research to meet the expectations of both reviewers and readers.
>
> **Q2**: Can we see the correlation for docking on ground truth and generated surface? What about the distance between ground truth and the generated surface in terms of their MaSIF distance?
>
> **A2**: We employ three evaluation metrics: the Intersection over Union (IoU) metric, which quantifies the overlap between the predicted and ground truth structures; the Chamfer distance, which measures the distance between the predicted and ground truth structures in terms of their surface normals; and Normal Consistency (NC), which assesses the consistency of surface normals between the predicted and ground truth structures. These three metrics collectively provide a comprehensive evaluation of the model's predictive capability from various perspectives.
>
> In addition to the evaluation metrics, we have provided some comparison figures between the ground truth and generated surfaces in the paper, such as Figure 3 and Figure 6. Furthermore, in the video available at https://anonymouswk.github.io/DSR, we showcase examples of the ground truth and reconstructed surfaces overlaid for comparison. These examples demonstrate the correlation between the ground truth and generated surfaces.
>
> Furthermore, we would like to address your concern about the MaSIF distance. While it is true that the MaSIF distance requires the full atomic coordinates, we have found that it is not directly applicable to our case. However, based on its calculation principle, the MaSIF distance is equivalent to our Chamfer distance. Therefore, we believe that our Chamfer distance metric is a suitable substitute for the MaSIF distance in this case.
>
> **Q3**: In addition, the author should consider spending more text to describe the evaluation setup, specifically around how train/test data is split and what has been done to prevent data leakage.
>
> **A3**: Thank you for your suggestion. We have incorporated additional details regarding the evaluation, including the splitting of datasets, in the manuscript.

---

> > ### Comment · Reviewer_2SKi · 2023-08-16
> >
> > Thank you for the additional experiments. I am raising my score to 7.

---

> > > ### Author Response · Authors · 2023-08-18
> > > **Response to Reviewer 2SKi**
> > >
> > > We appreciate your valuable and insightful comments.  We feel glad about your generally favorable assessment of our methodology.

---

### Author Rebuttal · Authors · 2023-08-09

I would like to extend my sincere appreciation for the reviewers’ efforts in coordinating the review process for our submission. In response to the reviewers' comments, we have carefully analyzed and compiled their concerns and suggestions. I would like to summarize the key points raised by the reviewers and assure you that we have taken them into serious consideration:

1. The reviewers mentioned the need for comparative experiments since we were the first to propose using surface representation to model protein dynamics, without any previous work for comparison. In response to the suggestion from Reviewer 2SKi, "Even though there is no prior work as a baseline, simple methods should still be used to confirm the learned model is not doing something trivial. For instance, what happens if the author just linearly interpolates the 3D coordinates for the experiment in 4.3?" we conducted additional experiments comparing our model with linear interpolation, and the results indeed demonstrated the effectiveness of our model.
2. Another concern raised by the reviewers was the generalization of our method and how it applies to simulating new proteins. Regarding this, I would like to clarify that it is indeed the focus of our future work. First and foremost, the reason we proposed using an implicit neural representation model to capture the dynamics of protein surfaces is because atoms and electrons have inherent uncertainties from the perspective of quantum mechanics. Representing proteins using atomic coordinates to calculate interactions may not always be accurate, while using electron density maps could be more appropriate. Therefore, we introduced surface representation. Our ultimate goal is to gradually achieve an alternative to traditional Molecular Dynamics (MD) simulations. However, the first step is to validate the feasibility of dynamic surface representation, which is the focus of our current study. In our model, we have also incorporated a latent code. Through this latent space, we aim to explore the representation space of proteins and enable the switching between different protein types by manipulating the latent code. Moving forward, our considerations include introducing vector field modeling and leveraging methods such as NeuralODE to achieve more realistic simulations in the following works.

In addition, the reviewers also pointed out several strengths and expressed a strong appreciation for our work. Specifically, they can be summarized as follows:

1. Application and novelty: Multiple reviewers mentioned that this paper presents an interesting direction in the field of protein surface modeling. Simulating protein dynamics using surface representation is considered a novel contribution, and the application of latent neural networks for modeling protein surfaces is seen as a novel approach.
2. Computational efficiency and practicality: The reviewers acknowledged the emphasis placed on the computational efficiency and practicality of our proposed method. They recognized the importance of learning efficient surrogate models for protein dynamics prediction, considering the computational cost associated with traditional molecular dynamics simulations. The reviewers also acknowledged the potential benefits of our approach for computational biology and drug discovery efforts, validating the significance of our work in their assessment.
3. Comprehensive analysis and presentation: The reviewers also commented on the quality of the writing in our paper. They noted that the introduction section effectively establishes the relationship with previous work and highlights the motivation behind the addressed task. Additionally, they appreciated the clear description of the datasets used for evaluation, which enhances reproducibility. Furthermore, they commended the extensive use of visuals, including images and videos in the supplementary materials, for effective visualization of the presented work.

In summary, the ultimate goal of our series of work is to model protein molecular dynamics using surface representation, which can also be applicable to downstream tasks. This paper serves as a starting point, aiming to validate the effectiveness of surface representation in modeling dynamics. This step is crucial and necessary for further progress. Furthermore, we have addressed all the reviewers' concerns and made corresponding modifications in the manuscript.

---

### Decision · Program_Chairs · 2023-09-21

**Decision:**

Accept (poster)

**Comment:**

This work proposes a new way to model protein surface dynamics through implicit neural networks. Reviewers agreed that the ideas are interesting and novel. Two key identified methodological limitations entail limited generalizability to new proteins and the current inability of the method to model the important surface biophysical properties. The paper is also lacking in terms of baselines. It would be valuable to complement the existing results with an investigation of how the proposed method compares to alternative approaches for predicting protein surface dynamics, e.g., by predicting how the entire structure evolves and retaining the surface. Additional ablations would also be insightful on which components of the method affect performance. Despite these limitations, the innovativeness of the method and problem make this paper worth presentation. The approach will hopefully inspire follow-up work from the authors and the community.